# ALIGNVLM: Bridging Vision and Language Latent Spaces for Multimodal Document Understanding

**Ahmed Masry**[1,2]**, Juan A. Rodriguez**[1,3,4]**, Tianyu Zhang**[1,3,5]**, Suyuchen Wang**[1,3,5]**,**
**Chao Wang**[1]**, Aarash Feizi**[1,3,6]**, Akshay Kalkunte Suresh**[1]**, Abhay Puri**[1]**,**
**Xiangru Jian**[1,7]**, Pierre-André Noël**[1]**, Sathwik Tejaswi Madhusudhan**[1]**,**
**Marco Pedersoli**[1,4]**, Bang Liu**[1,5,8]**, Nicolas Chapados**[1]**, Yoshua Bengio**[3,5,8]**,**
**Enamul Hoque**[2]**, Christopher Pal**[1,3,8,9]**, Issam H. Laradji**[1,10]**, David Vazquez**[1]**,**
**Perouz Taslakian**[1]**, Spandana Gella**[1]**, Sai Rajeswar**[1,3,5]

[1]ServiceNow    [2]York University    [3]Mila – Quebec AI Institute
[4]École de Technologie Supérieure    [5]Université de Montréal
[6]McGill University    [7]University of Waterloo    [8]CIFAR AI Chair
[9]Polytechnique Montréal    [10]University of British Columbia

## Abstract

Aligning visual features with language embeddings is a key challenge in vision-language models (VLMs). The performance of such models hinges on having a good connector that maps visual features generated by a vision encoder to a shared embedding space with the LLM while preserving semantic similarity. Existing connectors, such as multilayer perceptrons (MLPs), lack inductive bias to constrain visual features within the linguistic structure of the LLM's embedding space, making them data-hungry and prone to cross-modal misalignment. In this work, we propose a novel vision-text alignment method, ALIGNVLM, that maps visual features to a weighted average of LLM text embeddings. Our approach leverages the linguistic priors encoded by the LLM to ensure that visual features are mapped to regions of the space that the LLM can effectively interpret. ALIGNVLM is particularly effective for document understanding tasks, where visual and textual modalities are highly correlated. Our extensive experiments show that ALIGNVLM achieves state-of-the-art performance compared to prior alignment methods, with larger gains on document understanding tasks and under low-resource setups. We provide further analysis demonstrating its efficiency and robustness to noise.

## 1 Introduction

Vision-Language Models (VLMs) have gained significant traction in recent years as a powerful framework for multimodal document understanding tasks that involve interpreting both the visual and textual contents of scanned documents [Kim et al., 2022, Lee et al., 2023, Liu et al., 2023a, 2024, Hu et al., 2024, Wang et al., 2023a, Rodriguez et al., 2024b]. Such tasks are common in real-world commercial applications, including invoice parsing [Park et al., 2019], form reading [Jaume et al., 2019], and document question answering [Mathew et al., 2021b]. VLM architectures typically consist of three components: *(i)* a vision encoder to process raw images, *(ii)* a Large Language Model (LLM) pre-trained on text, and *(iii)* a connector module that maps the visual features from the vision encoder into the LLM's semantic space.

A central challenge in this pipeline is to effectively map the continuous feature embeddings of the vision encoder into the latent space of the LLM while preserving the semantic properties of visual concepts. Existing approaches can be broadly categorized into *deep fusion* and *shallow fusion* methods. *Deep fusion* methods, such as NVLM [Dai et al., 2024], Flamingo [Alayrac et al., 2022],

CogVLM [Wang et al., 2023b], and LLama 3.2-Vision [Grattafiori et al., 2024], integrate visual and textual features by introducing additional cross-attention and feed-forward layers at each layer of the LLM. While effective at enhancing cross-modal interaction, these methods substantially increase the parameter count of the VLM compared to the base LLM, resulting in high computational overhead and reduced efficiency.

In contrast, *shallow fusion* methods project visual features from the vision encoder into the LLM input embedding space using either multilayer perceptrons (MLPs) [Liu et al., 2023b, 2024], convolution mappings such as Honey-Bee [Cha et al., 2024] and H-Reducer [Hu et al., 2024], or attention-based mechanisms such as the Perceiver Resampler [Li et al., 2023b, Laurençon et al., 2024, Alayrac et al., 2022]. This approach is more parameter-efficient and computationally lighter than *deep fusion* method However, these connectors lack inductive bias to ensure that the projected features remain within the region spanned by the LLM's pretrained text embeddings. Consequently, the projected visual features may fall outside the distribution the LLM was trained on, leading to noisy or misaligned representations. Moreover, these mappings are typically learned from scratch, making them data-inefficient and less effective under low-resource conditions.

Recent methods like Ovis [Lu et al., 2024] attempt to alleviate these issues by introducing separate visual embeddings indexed from the vision encoder outputs and combined together

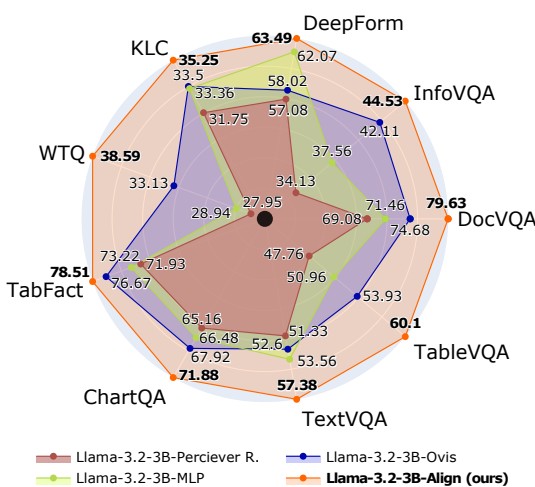

Figure 1: **Performance of Different VLM Connectors.** The proposed ALIGN connector outperforms other methods across benchmarks using the same training configuration. Radial distance is proportion of maximal score, truncated at 0.7 (black dot).

to construct the visual inputs to the LLM. However, this approach significantly increases parameter count due to the massive embedding matrix and requires extensive training to learn a new embedding space without guaranteeing alignment with the LLM's input latent space.

To address these limitations, this paper introduces **ALIGNVLM**, a novel framework that sidesteps direct projection of visual features into the LLM embedding space. Instead, our proposed connector, ALIGN, maps visual features into probability distributions over the LLM's *existing* pretrained vocabulary embeddings, which are then combined into a weighted representation of the text embeddings. By constraining each visual feature as a convex combination of the LLM text embeddings, our approach leverages the linguistic priors already encoded in the LLM's text space. This ensures that the resulting visual features lie within the convex hull of the LLM's embedding space, reducing the risk of noisy or out-of-distribution inputs and improving alignment between modalities. The connector thus enables faster convergence and stronger performance, particularly in low-resource scenarios.

Our experimental results show that ALIGN improves performance on various document understanding tasks, outperforming prior connector methods, with especially large gains in low-data regimes. We summarize our main contributions as follows:

- We propose a novel connector, ALIGN, to bridge the representation gap between vision and text modalities.

- We introduce a family of Vision-Language Models, **ALIGNVLM**, that achieves state-of-the-art performance on multimodal document understanding tasks by leveraging ALIGN.

- We conduct extensive experiments demonstrating the robustness and effectiveness of ALIGN across different LLM sizes and training data setups.

We release our code and research artifacts at alignvlm.github.io.

## 2 Related Work

### 2.1 Vision-Language Models

Over the past few years, Vision-Language Models (VLMs) have achieved remarkable progress, largely due to advances in Large Language Models (LLMs). Initially demonstrating breakthroughs in text understanding and generation [Brown et al., 2020, Raffel et al., 2023, Achiam et al., 2023, Grattafiori et al., 2024, Qwen et al., 2025, Team, 2024], LLMs are now increasingly used to effectively interpret visual inputs [Liu et al., 2023b, Li et al., 2024, Wang et al., 2024, Chen et al., 2024b, Dai et al., 2024, Drouin et al., 2024, Rodriguez et al., 2022]. This progress has enabled real-world applications across diverse domains, particularly in multimodal document understanding for tasks like form reading [Svetlichnaya, 2020], document question answering [Mathew et al., 2021b], and chart question answering [Masry et al., 2022]. VLMs commonly adopt a three-component architecture: a pretrained vision encoder [Zhai et al., 2023, Radford et al., 2021], a LLM, and a connector module. A key challenge for VLMs is effectively aligning visual features with the LLM's semantic space to enable accurate and meaningful multimodal interpretation.

### 2.2 Vision-Language Alignment for Multimodal Models

Existing vision-language alignment approaches can be classified into *deep fusion* and *shallow fusion*. Deep fusion methods integrate visual and textual features by modifying the LLM's architecture, adding cross-attention and feed-forward layers. For example, Flamingo [Alayrac et al., 2022] employs the Perceiver Resampler, which uses fixed latent embeddings to attend to vision features and fuses them into the LLM via gated cross-attention layers. Similarly, NVLM [Dai et al., 2024] adopts cross-gated attention while replacing the Perceiver Resampler with a simpler MLP. CogVLM [Wang et al., 2023b] extends this approach by incorporating new feed-forward (FFN) and QKV layers for the vision modality within every layer of the LLM. While these methods improve cross-modal alignment, they significantly increase parameter counts and computational overhead, making them less efficient.

On the other hand, shallow fusion methods are more computationally efficient, mapping visual features into the LLM's embedding space without altering its architecture. These methods can be categorized into three main types: *(1) MLP-based mapping*, such as LLaVA [Liu et al., 2023b] and PaliGemma [Beyer et al., 2024], which use multilayer perceptrons (MLP) to project visual features but often produce misaligned or noisy features due to a lack of constraints and inductive bias [Rodriguez et al., 2024b]; *(2) cross-attention mechanisms*, BLIP-2 [Li et al., 2023b] uses Q-Former, which utilizes a fixed set of latent embeddings to cross-attend to visual features, but that may still produce noisy or OOD visual features; *(3) convolution-based mechanisms*, such as HoneyBee [Cha et al., 2024] and H-Reducer [Hu et al., 2024], which leverage convolutional or ResNet [He et al., 2015] layers to preserve spatial locality while reducing dimensionality; and *(4) visual embeddings*, such as those introduced by Ovis [Lu et al., 2024], which use embeddings indexed by the vision encoder's outputs to produce the visual inputs. While this regularizes feature mapping, it adds substantial parameter overhead and creates a new vision embedding space, risking misalignment with the LLM's text embedding space. Encoder-free VLMs, like Fuyu-8B [1] and EVE [Diao et al., 2024], eliminate dedicated vision encoders but show degraded performance [Beyer et al., 2024].

In contrast, ALIGNVLM maps visual features from the vision encoder into probability distributions over the LLM's text embeddings, using them to compute a convex combination. By leveraging the linguistic priors encoded in the LLM's vocabulary, ALIGNVLM ensures that visual features remain within the convex hull of the text embedding. This design mitigates noisy or out-of-distribution projections and achieves stronger multimodal alignment, particularly in tasks that require joint modalities representation like multimodal document understanding and in low-resource settings.

## 3 Methodology

### 3.1 Model Architecture

The overall model architecture, shown in Figure 2, consists of three main components:

---

[1]https://www.adept.ai/blog/fuyu-8b

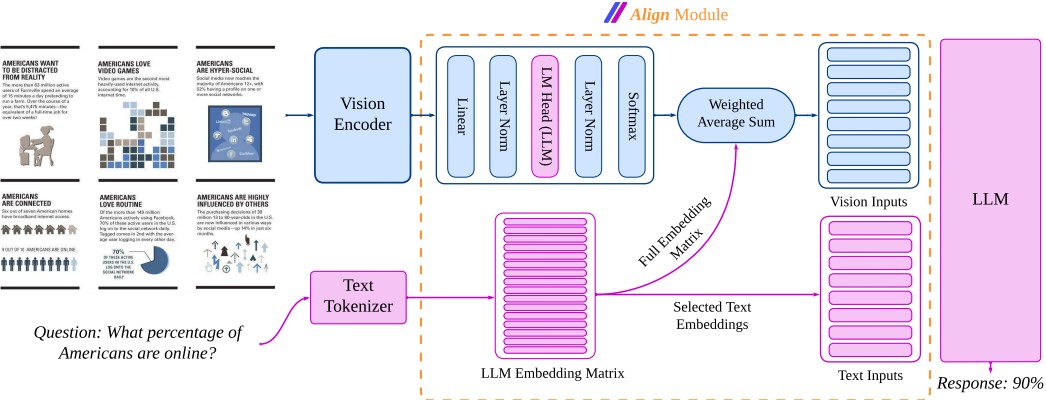

Figure 2: **ALIGNVLM Model Architecture.** The vision encoder extracts image features, which are processed to produce probabilities over the LLM embeddings. A weighted average combines these probabilities with embeddings to generate vision input vectors. Text inputs are tokenized, and the corresponding embeddings are selected from the embedding matrix, which is then used as input to the LLM. We display the vision layers in blue, and the text layers in purple.

**(1) Vision Encoder.** To handle high-resolution images of different aspect ratios, we divide each input image into multiple tiles according to one of the predefined aspect ratios (e.g., 1:1, 1:2, ..., 9:1) chosen via a coverage ratio [Lu et al., 2024, Chen et al., 2024a]. Due to limited computational resources, we set the maximum number of tiles to 9. Each tile is further partitioned into $14 \times 14$ patches, projected into vectors, and processed by a SigLip-400M vision encoder [Zhai et al., 2023] to extract contextual visual features.

Each tile $t \in \{1, \cdots, T\}$ is divided into $N_t$ patches

$$\mathbf{P}_t = \{\mathbf{p}_{t,1}, \cdots, \mathbf{p}_{t,N_t}\},$$

where $\mathbf{p}_{t,i}$ is the $i$-th patch of tile $t$. The vision encoder maps these patches to a set of visual feature vectors

$$\mathbf{F}_t = \text{VisionEncoder}(\mathbf{P}_t), \quad \mathbf{F}_t = \{\mathbf{f}_{t,1}, \cdots, \mathbf{f}_{t,N_t}\}, \quad \mathbf{f}_{t,i} \in \mathbb{R}^d.$$

Finally, we concatenate the feature sets across all tiles into a single output

$$\mathbf{F} = \text{concat}\Big(\mathbf{F}_1, \mathbf{F}_2, \cdots, \mathbf{F}_T\Big).$$

**(2) ALIGN Module.** This module aligns the visual features with the LLM. A linear layer $\mathbf{W}_1 \in \mathbb{R}^{D \times d}$ first projects the visual features $\mathbf{F} \in \mathbb{R}^{T \cdot N_t \times d}$ to the LLM's token embedding space: one $\mathbb{R}^D$ vector per token. A second linear layer $\mathbf{W}_2 \in \mathbb{R}^{V \times D}$ (initialized from the LLM's language-model head) followed by a softmax, produces a probability simplex $\mathbf{P}_{\text{vocab}}$ over the LLM's vocabulary ($V$ tokens)

$$\mathbf{P}_{\text{vocab}} = \text{softmax}(\text{LayerNorm}(\mathbf{W}_2 \, \text{LayerNorm}(\mathbf{W}_1 \mathbf{F}))) \tag{1}$$

We then use the LLM text embeddings $\mathbf{E}_{\text{text}} \in \mathbb{R}^{V \times D}$ to compute a weighted sum

$$\mathbf{F}'_{\text{align}} = \mathbf{P}_{\text{vocab}}^\top \mathbf{E}_{\text{text}}. \tag{2}$$

Finally, we concatenate $\mathbf{F}'_{\text{align}}$ with the tokenized text embeddings to form the LLM input

$$\mathbf{H}_{\text{input}} = \text{concat}\big(\mathbf{F}'_{\text{align}}, \mathbf{E}_{\text{text}}(\mathbf{x})\big),$$

where $\mathbf{E}_{\text{text}}(\mathbf{x})$ is obtained by tokenizing the input text $\mathbf{x} = (x_1, \cdots, x_M)$ and selecting the corresponding embeddings from $\mathbf{E}_{\text{text}}$ such that

$$\mathbf{E}_{\text{text}}(\mathbf{x}) = \big[\mathbf{E}_{\text{text}}(x_1), \cdots, \mathbf{E}_{\text{text}}(x_M)\big]. \tag{3}$$

**(3) Large Language Model.** We feed the concatenated vision and text vectors, $\mathbf{H}_{\text{input}}$, into the LLM, which then generates output text auto-regressively. To demonstrate the effectiveness of our alignment technique, we experiment with the Llama 3.1 model family [Grattafiori et al., 2024]. These models offer state-of-the-art performance and permissive licenses, making them suitable for commercial applications. In particular, we utilize Llama 3.2-1B, Llama 3.2-3B, and Llama 3.1-8B.

## 3.2 Motivation and relation with existing methods

By construction, each $\mathbb{R}^D$ representation in $\mathbf{F}'_{\text{align}}$ is constrained to the convex hull of the points $\mathbb{E}_{\text{text}}$, thus concentrating the visual features in the part of latent space that the LLM can effectively interpret. Moreover, we argue that our initialization of $\mathbf{W}_2$ to the language model head is an inductive bias toward *recycling* some of the semantics of these text tokens into visual tokens. This contrasts with past methods that have been proposed to adapt the vision encoder outputs $\mathbf{F} \in \mathbb{R}^{T \cdot N_t \times d}$ to an $\mathbf{F}' \in \mathbb{R}^{T \cdot N_t \times D}$ to be fed to the LLM. Here, we consider two examples in more detail, highlighting these contrasts.

*(1) MLP Connector* Liu et al. [2023b] applies a linear projection with parameters $\mathbf{W}_{\text{MLP}} \in \mathbb{R}^{D \times d}$ and $\mathbf{b}_{\text{MLP}} \in \mathbb{R}^D$, followed by an activation function $\sigma$ (e.g., ReLU)

$$\mathbf{F}'_{\text{MLP}} = \sigma(\mathbf{W}_{\text{MLP}}\mathbf{F} + \mathbf{b}_{\text{MLP}}).$$

These parameters are all learned from scratch, without any bias aligning them to text embeddings.

*(2) Visual Embedding Table* Lu et al. [2024] introduces an entire new set of visual embeddings $\mathbf{E}_{\text{VET}} \in \mathbb{R}^{K \times D}$ which, together with the weights $\mathbf{W}_{\text{VET}} \in \mathbb{R}^{K \times d}$, specifies

$$\mathbf{F}'_{\text{VET}} = \text{softmax}(\mathbf{W}_{\text{VET}}\mathbf{F})^\top \mathbf{E}_{\text{VET}}.$$

When $D < d$, our $\mathbf{W}_2\mathbf{W}_1$ amounts to a low-rank version of $\mathbf{W}_{\text{VET}}$. There is thus much more to learn to obtain $\mathbf{F}'_{\text{VET}}$, and there is again no explicit pressure to align it with the text embeddings.

## 3.3 Training Datasets & Stages

We train our model in three stages:

**Stage 1.** This stage focuses on training the ALIGN Module to map visual features to the LLM's text embeddings effectively. We use the CC-12M dataset Changpinyo et al. [2021], a large-scale web dataset commonly used for VLM pretraining Liu et al. [2023b], which contains 12M image-text pairs. However, due to broken or unavailable links, we retrieved 8.1M pairs. This dataset facilitates the alignment of visual features with the text embedding space of the LLM. During this stage, we train the full model, as this approach improves performance and stabilizes the ALIGN Module training.

**Stage 2.** The goal is to enhance the model's document understanding capabilities, such as OCR, document structure comprehension, in-depth reasoning, and instruction-following. We leverage the BigDocs-7.5M dataset Rodriguez et al. [2024a], a curated collection of license-permissive datasets for multimodal document understanding. This dataset aligns with the Accountability, Responsibility, and Transparency (ART) principles Bommasani et al. [2023], Vogus and Llansóe [2021], ensuring compliance for commercial applications. As in Stage 1, we train the full model during this stage.

**Stage 3.** To enhance the model's instruction-tuning capabilities, particularly for downstream tasks like question answering, we further train it on the DocDownstream Rodriguez et al. [2024a], Hu et al. [2024] instruction tuning dataset. In this stage, the vision encoder is frozen, focusing training exclusively on the LLM and ALIGN module.

# 4 Experimental Setup

**Setup.** We conduct all experiments using 8 nodes of H100 GPUs, totaling 64 GPUs. For model training, we leverage the MS-Swift framework [Zhao et al., 2024] for its flexibility. Additionally, we utilize the DeepSpeed framework [Aminabadi et al., 2022], specifically the ZeRO-3 configuration, to optimize efficient parallel training across multiple nodes. Detailed hyperparameters are outlined in Appendix A.1.

Table 1: **Main Results on General Document Benchmarks.** We compare ALIGNVLM (ours) with state-of-the-art (SOTA) open and closed-source instructed models, and with base models that we trained using the process described in Section 3.3. ALIGNVLM models outperform all Base VLM models trained in the same data regime. Our models also perform competitively across document benchmarks even compared with SOTA models, in which the data regime is more targeted and optimized. Color coding for comparison: closed-source models, open-source models below 7B parameters, open-source models between 7-12B parameters.

| Model | DocVQA VAL | InfoVQA VAL | DeepForm TEST | KLC TEST | WTQ TEST | TabFact TEST | ChartQA TEST | TextVQA VAL | TableVQA TEST | Avg. Score |
|---|---|---|---|---|---|---|---|---|---|---|
| **Closed-Source VLMs** *(Opaque Training Data)* | | | | | | | | | | |
| Claude-3.5 Sonnet | 88.48 | 59.05 | 31.41 | 24.82 | 47.13 | 53.48 | 51.84 | **71.42** | **81.27** | 56.54 |
| GeminiPro-1.5 | 91.23 | **73.94** | 32.16 | 24.07 | **50.29** | 71.22 | 34.68 | 68.16 | 80.43 | 58.46 |
| GPT-4o 20240806 | **92.80** | 66.37 | **38.39** | 29.92 | 46.63 | **81.10** | **85.70** | 70.46 | 72.87 | **64.91** |
| **Open-Source Instruct VLMs** *(Semi-Opaque Training Data)* | | | | | | | | | | |
| Janus-**1.3B** [Wu et al., 2024a] | 30.15 | 17.09 | 0.62 | 15.06 | 9.30 | 51.34 | 57.20 | 51.97 | 18.67 | 27.93 |
| Qwen2-VL-**2B** [Wang et al., 2024] | 89.16 | 64.11 | 32.38 | 25.18 | 38.20 | 57.21 | 73.40 | 79.90 | 43.07 | 55.84 |
| Qwen2.5-VL-**3B** [Wang et al., 2024] | **93.00** | **75.83** | 32.84 | 24.82 | **53.46** | **71.16** | **83.91** | 79.29 | **71.66** | **65.10** |
| InternVL-2.5-**2B** [Chen et al., 2024b] | 87.70 | 61.85 | 13.14 | 16.58 | 36.33 | 57.26 | 74.96 | 76.85 | 42.20 | 51.87 |
| InternVL-3-**2B** [Zhu et al., 2025] | 87.33 | 66.99 | **37.90** | **29.79** | 39.44 | 59.91 | 75.32 | 78.69 | 43.46 | 57.64 |
| DeepSeek-VL2-Tiny-**3.4B** [Wu et al., 2024b] | 88.57 | 63.86 | 25.11 | 19.04 | 35.07 | 52.15 | 80.92 | **80.48** | 56.30 | 55.72 |
| Phi3.5-Vision-**4B** [Abdin et al., 2024] | 86.00 | 56.20 | 10.47 | 7.49 | 17.18 | 30.43 | 82.16 | 73.12 | 70.70 | 48.19 |
| Qwen2-VL-**7B** [Wang et al., 2024] | 93.83 | 76.12 | 34.55 | 23.37 | 52.52 | 74.68 | 83.16 | 84.48 | 53.97 | 64.08 |
| Qwen2.5-VL-**7B** [Bai et al., 2025] | **94.88** | **82.49** | 42.21 | 24.26 | **61.96** | 78.56 | **86.00** | **85.35** | 76.10 | **70.20** |
| LLaVA-NeXT-**7B** [Xu et al., 2024] | 63.51 | 30.90 | 1.30 | 5.35 | 20.06 | 52.83 | 52.12 | 65.10 | 32.87 | 36.00 |
| DocOwl1.5-**8B** [Hu et al., 2024] | 80.73 | 49.94 | **68.84** | **37.99** | 38.87 | **79.67** | 68.56 | 68.91 | 52.60 | 60.68 |
| InternVL-2.5-**8B** [Chen et al., 2024b] | 91.98 | 75.36 | 34.55 | 22.31 | 50.33 | 74.75 | 82.84 | 79.00 | 52.10 | 62.58 |
| InternVL-3-**8B** [Zhu et al., 2025] | 91.99 | 73.90 | 51.24 | 36.41 | 53.60 | 72.27 | 85.60 | 82.41 | 53.26 | 66.74 |
| Fuyu-**8B** [Bavishi et al., 2023] | 48.97 | 23.09 | 4.78 | 6.63 | 14.55 | 47.91 | 44.36 | 46.02 | 15.49 | 22.97 |
| Ovis-1.6-Gemma2-**9B** [Lu et al., 2024] | 88.84 | 73.97 | 45.16 | 23.91 | 50.72 | 76.66 | 81.40 | 77.73 | 48.33 | 62.96 |
| Llama3.2-**11B** [Grattafiori et al., 2024] | 82.71 | 36.62 | 1.78 | 3.47 | 23.03 | 58.33 | 23.80 | 54.28 | 22.40 | 34.04 |
| Pixtral-**12B** [Agrawal et al., 2024] | 87.67 | 49.45 | 27.37 | 24.07 | 45.18 | 73.53 | 71.80 | 76.09 | 67.13 | 58.03 |
| **Document Understanding Instructed Models** *(Instruction Tuned on BigDocs-7.5M + DocDownStream [Rodriguez et al., 2024a, Hu et al., 2024])* | | | | | | | | | | |
| Qwen2-VL-2B (base+) [Wang et al., 2024] | 57.23 | 31.88 | 49.31 | 34.39 | 31.61 | 64.75 | 68.60 | **61.01** | 47.53 | 49.59 |
| ALIGNVLM-Llama-3.2-**1B** (ours) | 72.42 | 38.16 | 60.47 | 33.71 | 33.71 | 65.44 | 48.81 | 50.29 | 52.14 | |
| ALIGNVLM-Llama-3.2-**3B** (ours) | **79.63** | **44.53** | **63.49** | 35.25 | 38.59 | **78.51** | **71.88** | 57.38 | 60.10 | **58.81** |
| DocOwl1.5-**8B** (base+) [Hu et al., 2024] | 78.70 | 47.62 | 64.39 | 36.93 | 35.69 | 72.65 | 65.80 | 67.30 | 49.03 | 57.56 |
| Llama3.2-**11B** (base+) [Grattafiori et al., 2024] | 78.99 | 44.27 | **67.05** | **37.22** | 40.18 | 78.04 | 71.40 | **68.46** | 56.73 | 60.26 |
| ALIGNVLM-Llama-3.1-**8B** (ours) | **81.18** | **53.75** | 63.25 | 35.50 | **45.31** | **83.04** | 75.00 | 64.60 | **64.33** | **62.88** |

**Baselines.** Our work focuses on architectural innovations, so we ensure that all baselines are trained on the same datasets. To enable fair comparisons, we evaluate our models against a set of **Base VLMs** fine-tuned on the same instruction-tuning tasks (Stages 2 and 3) as our models, using the BigDocs-7.5M and BigDocs-DocDownstream datasets. This approach ensures consistent training data, avoiding biases introduced by the **Instruct** versions of VLMs, which are often trained on undisclosed instruction-tuning datasets. Due to the scarcity of recently released publicly available Base VLMs, we primarily compare our model against the following Base VLMs of varying sizes: Qwen2-VL-2B [Wang et al., 2024], DocOwl1.5-8B [Hu et al., 2024], and LLama 3.2-11B [Grattafiori et al., 2024].

For additional context, we also include results from the Instruct versions of recent VLMs of different sizes: Phi3.5-Vision-4B [Abdin et al., 2024], Qwen2-VL-2B and 7B [Wang et al., 2024], Qwen2.5-VL-7B [Qwen et al., 2025], LLaVA-NeXT-7B [Liu et al., 2024], InternVL2.5-2B and 8B [Chen et al., 2024b], InternVL3-2B and 8B [Zhu et al., 2025], Janus-1.3B [Wu et al., 2024a], DeepSeek-VL2-Tiny [Wu et al., 2024b], Ovis1.6-Gemma-9B [Lu et al., 2024], Llama3.2-11B [Grattafiori et al., 2024], DocOwl1.5-8B [Hu et al., 2024], and Pixtral-12B [Agrawal et al., 2024].

**Evaluation Benchmarks.** We evaluate our models on a diverse range of document understanding benchmarks that assess the model's capabilities in OCR, chart reasoning, table processing, or form comprehension. In particular, we employ the VLMEvalKit [Duan et al., 2024] framework and report the results on the following popular benchmarks: DocVQA [Mathew et al., 2021b], InfoVQA [Mathew et al., 2021a], DeepForm [Svetlichnaya, 2020], KLC [Stanisławek et al., 2021], WTQ [Pasupat and Liang, 2015], TabFact [Chen et al., 2020], ChartQA [Masry et al., 2022], TextVQA [Singh et al., 2019], and TableVQA [Kim et al., 2024].

Table 2: **Impact of Connector Designs on VLM Performance:** We present the results of experiments evaluating different connector designs for conditioning LLMs on visual features. Our proposed **ALIGN** connector is compared against a basic Multi-Layer Perceptron (**MLP**), the **Perceiver Resampler**, and **Ovis**. The results demonstrate that ALIGN consistently outperforms these alternatives across all benchmarks.

| Model | DocVQA VAL | InfoVQA VAL | DeepForm TEST | KLC TEST | WTQ TEST | TabFact TEST | ChartQA TEST | TextVQA VAL | TableVQA TEST | Avg. Score |
|---|---|---|---|---|---|---|---|---|---|---|
| Llama-3.2-3B-**MLP** | 71.46 | 37.56 | 62.07 | 33.36 | 28.94 | 73.22 | 66.48 | 53.56 | 50.96 | 53.06 |
| Llama-3.2-3B-**Perciever R.** | 69.08 | 34.13 | 57.08 | 31.75 | 27.95 | 71.93 | 65.16 | 51.33 | 47.76 | 50.68 |
| Llama-3.2-3B-**Ovis** | 74.68 | 42.11 | 58.02 | 33.50 | 33.13 | 76.67 | 67.92 | 52.60 | 53.93 | 54.72 |
| Llama-3.2-3B-**ALIGN** (ours) | **79.63** | **44.53** | **63.49** | **35.25** | **38.59** | **78.51** | **71.88** | **57.38** | **60.10** | **58.81** |

# 5 Results

## 5.1 Main Results

Table 1 presents the performance of ALIGNVLM compared to state-of-the-art (SOTA) open- and closed-source instructed models, as well as baseline Base VLMs fine-tuned in the same instruction-tuning setup. The results demonstrate that ALIGNVLM consistently outperforms all Base VLMs within the same size category and achieves competitive performance against SOTA Instruct VLMs despite being trained on a more limited data regime. Below, we provide a detailed analysis.

**ALIGNVLM vs. Base VLMs.** Our ALIGNVLM models, based on Llama 3.2-1B and Llama 3.2-3B, significantly outperform the corresponding Base VLM, Qwen2-VL-2B, by up to *9.22%*. Notably, ALIGNVLM-Llama-3.2-3B surpasses DocOwl1.5-8B, which has 4B more parameters, demonstrating the effectiveness of ALIGN in enhancing multimodal capabilities compared to traditional *shallow fusion* methods (e.g., MLPs). Furthermore, our 8B model achieves a *2.62%* improvement over Llama3.2-11B despite sharing the same Base LLM, Llama3.1-8B. Since all models in this comparison were trained on the same instruction-tuning setup, this experiment provides a controlled evaluation, isolating the impact of architectural differences rather than dataset biases. Consequently, these results suggest that ALIGNVLM outperforms VLMs with shallow fusion techniques and surpasses parameter-heavy *deep fusion* VLMs, such as Llama3.2-11B, while maintaining a more efficient architecture.

**ALIGNVLM vs. Instruct VLMs.** Even as open-source Instruct models are trained on significantly larger, often undisclosed instruction-tuning datasets, ALIGNVLM achieves competitive performance. For example, ALIGNVLM-Llama-3.2-3B (*58.81%*) outperforms other strong instruction-tuned VLMs in its size class, such as Qwen2-VL-2B and InternVL-3-2B, by considerable margins (*2.97%* and *1.17%*, respectively). While it falls slightly behind Qwen2.5-VL-3B, a direct comparison is not entirely fair, as the latter was trained on a proprietary instruction-tuning dataset.

Additionally, our 8B model outperforms significantly larger models such as Llama 3.2-11B and PixTral-12B by substantial margins. It also surpasses InternVL-2.5-8B and performs competitively with Qwen2.5-VL-7B, though a direct comparison may not be entirely fair since Qwen2.5-VL-7B was trained on an undisclosed instruction-tuning dataset. Finally, ALIGNVLM also exhibits comparable performance to closed-source models like GeminiPro-1.5 and GPT4o.

Overall, these results validate the effectiveness of ALIGN and establish ALIGNVLM as a state-of-the-art model for multimodal document understanding.

## 5.2 Impact of Connector Designs on VLM Performance

### 5.2.1 High-Resource Training Regime

To assess the effectiveness of our ALIGN module, we compare it against three different and widely used *shallow fusion* VLM connectors: MLP, Perceiver Resampler, and Ovis. These experiments were carefully conducted under precisely identical training conditions (datasets, hyperparameters, training stages) as outlined in Appendix A.1, ensuring a fair and rigorous comparison. The results in Table 2 show that ALIGN consistently outperforms all alternatives, demonstrating its superiority

Table 3: **Connector Performance under a Low-Resource Training Regime:** We evaluate the effectiveness of more shallow-fusion connectors when trained on limited data. The ALIGN connector achieves the highest performance, with notably larger gains on document understanding tasks, demonstrating its data efficiency and strong inductive bias.

| Model | Document Understanding Tasks | | | | | General Vision Tasks | | | | | |
|---|---|---|---|---|---|---|---|---|---|---|---|
| | DocVQA | InfoVQA | ChartQA | TextVQA | Avg. | MMMU | SeedBench | MMVet | POPE | GQA | Avg. |
| Llama-3.2-3B-MLP | 42.11 | 19.93 | 48.44 | 51.97 | 40.61 | 33.33 | 58.54 | 31.14 | 87.35 | 57.62 | 53.59 |
| Llama-3.2-3B-Perceiver | 32.18 | 18.10 | 40.00 | 44.31 | 33.64 | 35.22 | **63.70** | 26.19 | 84.92 | 55.86 | 53.17 |
| Llama-3.2-3B-Ovis | 57.73 | 26.39 | 54.52 | 55.60 | 48.56 | 31.89 | 60.97 | 30.41 | 88.26 | 56.23 | 53.55 |
| Llama-3.2-3B-Hreducer | 34.59 | 17.57 | 45.64 | 47.13 | 36.23 | 35.00 | 61.82 | 28.39 | 87.48 | 58.24 | 54.18 |
| Llama-3.2-3B-HoneyBee | 55.86 | 19.36 | 55.32 | 58.13 | 47.16 | 32.11 | 61.18 | 34.31 | **89.28** | 54.79 | 54.33 |
| Llama-3.2-3B-ALIGN (ours) | **71.43** | **30.50** | **69.72** | **65.63** | **59.32** | **35.33** | 63.27 | **35.32** | 88.85 | **61.67** | **56.88** |

both in aligning visual and textual modalities in multimodal document understanding. MLP and Perceiver Resampler achieve the lowest performance, 53.06% and 50.68%, respectively, due to their direct feature projection, which lacks an explicit mechanism to align visual features with the LLM's text space, leading to misalignment. Ovis introduces a separate visual embedding table, but this additional complexity does not significantly improve alignment, yielding only 54.72% accuracy. In contrast, ALIGN ensures that visual features remain within the convex hull of the LLM's text latent space, leveraging the linguistic priors of the LLM to enhance alignment and mitigate noisy embeddings. This design leads to the highest performance (58.81%), establishing ALIGN as the most effective connector for integrating vision and language in multimodal document understanding. We provide some example outputs of the Llama-3.2-3B models with different connector designs in Appendix A.4. Furthermore, we include an analysis of the runtime efficiency and memory usage of different connectors in Appendix A.2.

### 5.2.2 Low-Resource Training Regime

The previous section focused on large-scale training setups involving millions of data samples (BigDocs-7.5M), which require significant compute resources and limit the number of baselines that we were able to compare against. Here, we examine whether ALIGN remains effective in a *low-resource* setting.

We conduct additional experiments using SigLIP-400M as the vision encoder and Llama-3.2-3B as the language model, fine-tuned on the LLaVA-NeXT dataset Liu et al. [2024], which contains *779K* samples. We follow the official LLaVA-NeXT configuration for both training stages. *(i) Pretraining:* the model is trained on the LLaVA-558K image–caption dataset Liu et al. [2024], freezing both the LLM and vision encoder while fine-tuning the connector (learning rate = 1e-3, batch size = 32, 1 epoch on 8 × H100 GPUs). To handle high-resolution document images, we adopt the "anyres_max_9" strategy with grid weaving from 1×1 to 6×6, supporting resolutions up to 2304×2304 with 729 tokens per grid; *(ii) Instruction tuning:* the model is further fine-tuned on the LLaVA-NeXT-779K instruction dataset with learning rates of 1e-5 for the LLM and connector, 2e-6 for the vision encoder, batch size = 8, for 1 epoch.

This lightweight setup allows direct comparison across more connector architectures including MLP Liu et al. [2023a], Perceiver Resampler, Ovis Lu et al. [2024], H-Reducer (1×4) Hu et al. [2024], and HoneyBee (C-Abstractor) Cha et al. [2024], all trained under identical conditions for fairness. Since the LLaVA-Next dataset is general-purpose and not exclusively document-focused like BigDocs-7.5M [Rodriguez et al., 2024a], it allows us to evaluate whether the ALIGN connector generalizes beyond document understanding to broader visual reasoning. Accordingly, we assess all models on a comprehensive suite of benchmarks spanning both document understanding and general vision–language tasks. The document understanding benchmarks include DocVQA Mathew et al. [2021b], InfoVQA Mathew et al. [2021a], ChartQA Masry et al. [2022], and TextVQA Singh et al. [2019]. For general vision–language evaluation, we report results on MMMU-dev Yue et al. [2024], SeedBench Li et al. [2023a], and MMVet Yu et al. [2024], Pope [Li et al., 2023c], and GQA [Hudson and Manning, 2019].

As summarized in Table 3, ALIGN consistently outperforms other connectors under this low-data regime, with stronger gains on document understanding tasks. The wider performance margin

Table 4: Performance comparison when evaluating ALIGN with the full text embedding vocabulary (128K) versus the reduced subset of 3.4K high-probability embeddings. The results show negligible performance degradation, indicating that ALIGN relies primarily on a small subset of embeddings.

| Model | DocVQA VAL | InfoVQA VAL | DeepForm TEST | KLC TEST | WTQ TEST | TabFact TEST | ChartQA TEST | TextVQA VAL | TableVQA TEST | Avg. Score |
|---|---|---|---|---|---|---|---|---|---|---|
| Llama-3.2-3B-ALIGN (**Full Embeddings**) | 79.63 | 44.53 | 63.49 | 35.25 | 38.59 | 78.51 | 71.88 | 57.38 | 60.10 | 58.81 |
| Llama-3.2-3B-ALIGN (**3.4K Embeddings**) | 79.40 | 44.13 | 63.64 | 35.02 | 38.26 | 78.83 | 71.72 | 57.48 | 59.80 | 58.69 |

between ALIGN and others connectors under limited data (Table 3) compared to the high-resource setting (Table 2) underscores the benefit of its inductive bias. By grounding visual features within the LLM's text embedding space, ALIGN learns more efficiently from fewer samples, unlike direct-projection connectors that rely heavily on large datasets. This makes ALIGN especially valuable for resource-constrained environments such as academic labs or small-scale industrial research setups, where both data and compute are limited.

## 5.3 Probability Distribution over Text Tokens Analysis

To better understand the behavior of ALIGN, we examine the probability distribution, $\mathbf{P}_{\text{vocab}}$ in Eq (1), over the LLM's text vocabulary generated from visual features. Specifically, we process 100 document images through the vision encoder and ALIGN, then average the resulting probability distributions across all image patches. The final distribution is shown in Figure 3. As illustrated, the distribution is *dense* (rather than sparse), with the highest probability assigned to a single token being *0.0118*. This can be explained by the vision feature space being continuous and of much higher cardinality than the discrete text space. Indeed, while the LLM has 128K distinct vocabulary tokens, an image patch (e.g., 14×14 pixels) contains continuous, high-dimensional information that cannot be effectively mapped to a single or a few discrete tokens.

We conducted a deeper analysis of the token probability distributions produced by the ALIGN connector. Our observations show that ALIGN consistently assigns high probabilities to approximately 3.4K tokens from the entire vocabulary, while the remaining tokens receive negligible probabilities (below $10^{-6}$). To better understand this behavior, we applied Principal Component Analysis (PCA) to reduce the dimensionality of the embeddings and visualized them in a two-dimensional space, as shown in Figure 4. The visualization reveals that these 3.4K tokens densely and comprehensively span the latent space of the LLM's text embeddings. To validate this finding, we conducted additional evaluation experiments in which we retained only these 3.4K high-probability embeddings in the ALIGN connector, entirely removing the rest during evaluation. As shown in Table 4, the performance difference compared to using the full embedding set (128K) was negligible. This confirms that ALIGN effectively leverages and combines a compact subset of embeddings to map visual features into semantically meaningful regions within the LLM's latent text space. Moreover, this suggests that ALIGN can be further optimized through targeted embedding pruning to improve computational efficiency without sacrificing performance.

## 5.4 Robustness to Noise Analysis

To evaluate the robustness of our **ALIGN** connector to noisy visual features, we conduct an experiment where random Gaussian noise is added to the visual features produced by the vision encoder before passing them into the connector. Specifically, given the visual features $\mathbf{F} \in \mathbb{R}^{N \times d}$ output by the vision encoder (where $N$ is the number of feature vectors and $d$ is their dimensionality), we perturbed them as

$$\widetilde{\mathbf{F}} = \mathbf{F} + \mathbf{N}, \quad \mathbf{N} \sim \mathcal{N}(0, \sigma = 3).$$

Table 5: **Robustness to Noise.** Comparison of Avg. Scores with and without Gaussian noise ($\sigma = 3$), including performance drop ($\Delta$).

| Model | Without Noise | With Noise | Drop ($\Delta$) |
|---|---|---|---|
| Llama-3.2-3B-MLP | 53.06 | 27.52 | ↓ 25.54 |
| Llama-3.2-3B-ALIGN (ours) | **58.81** | **57.14** | ↓ **1.67** |

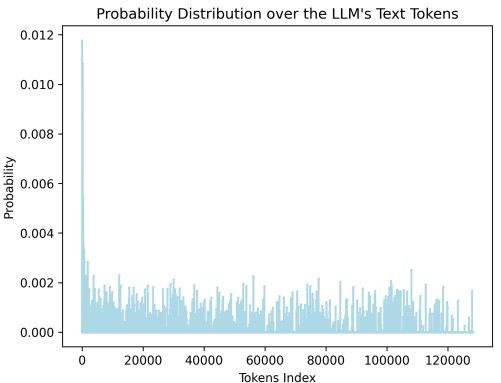

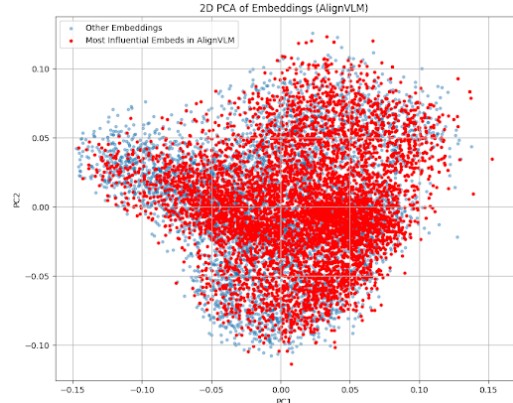

Figure 3: **Probability distribution over LLM tokens**, highlighting dense probabilities for whitespace tokens.

Figure 4: **PCA of ALIGN Embeddings:** The principal components of the most influential embeddings in the Align Connector span most of the feature space represented by all embeddings.

As shown in Table 5, our **ALIGN** connector demonstrates high robustness to noise, with only a *1.67%* average drop in performance. In contrast, the widely adopted MLP connector suffers a significant performance degradation of *25.54%*, highlighting its vulnerability to noisy inputs. Furthermore, we measured the average cosine distance between the original and noise-perturbed visual embeddings using both the ALIGN and MLP connectors. ALIGN showed significantly lower distances (0.0036) than MLP (0.3938), further validating its robustness to noise. These empirical results support our hypothesis that leveraging the knowledge encoded in the LLM's text embeddings and constraining the visual features within the convex hull of the text latent space act as a regularization mechanism, reducing the model's sensitivity to noisy visual features.

## 6 Conclusion

We introduce ALIGN, a novel connector designed to align vision and language latent spaces in vision-language models (VLMs), specifically enhancing multimodal document understanding. By improving cross-modal alignment and minimizing noisy embeddings, our models, ALIGNVLM, which leverage ALIGN, achieve state-of-the-art performance across diverse document understanding tasks. This includes outperforming base VLMs trained on the same datasets and achieving competitive performance with open-source instruct models trained on undisclosed data. Extensive experiments and ablations validate the robustness and effectiveness of ALIGN compared to existing connector designs, establishing it as a significant contribution to vision-language modeling. Future work will explore training on more diverse instruction-tuning datasets to generalize to broader domains.

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

# A Appendix

## A.1 Experimental Setup

We provide detailed hyperparameters of our experiments in Table 6.

Table 6: Detailed hyperparameters for each training stage across different LLM backbones.

| LLM Backbone | Llama 3.2-1B | | | Llama 3.2-3B | | | Llama 3.1-8B | | |
|---|---|---|---|---|---|---|---|---|---|
| | Stage-1 | Stage-2 | Stage-3 | Stage-1 | Stage-2 | Stage-3 | Stage-1 | Stage-2 | Stage-3 |
| Trainable Parameters | Full Model | Full Model | LLM & Connector | Full Model | Full Model | LLM & Connector | Full Model | Full Model | LLM & Connector |
| Batch Size | 512 | 512 | 512 | 512 | 256 | 256 | 512 | 256 | 256 |
| Text Max Length | 1024 | 2048 | 2048 | 1024 | 2048 | 2048 | 1024 | 2048 | 2048 |
| Epochs | 1 | 1 | 5 | 1 | 1 | 5 | 1 | 1 | 5 |
| Learning Rate | $1 \times 10^{-5}$ | $5 \times 10^{-5}$ | $5 \times 10^{-5}$ | $1 \times 10^{-5}$ | $5 \times 10^{-5}$ | $5 \times 10^{-5}$ | $1 \times 10^{-5}$ | $1 \times 10^{-5}$ | $1 \times 10^{-5}$ |

## A.2 Runtime Comparison Between Connectors

One caveat in the ALIGN connector is that it includes an additional LM head layer, which slightly increases the total number of parameters. However, this addition has a negligible impact on runtime efficiency due to its simple structure. It only introduces a few matrix multiplication operations (as shown in Equations 1 and 2) instead of stacking many complex layers that require sequential processing, as in deep fusion methods.

To empirically validate this claim, we benchmarked the runtime and memory usage of models equipped with different connector types (MLP, Align, Ovis, and Perceiver), following the same experimental setup as in Table 2. As shown in Table 7, the results demonstrate that although the ALIGN connector delivers notably superior performance (see Table 2), the variations in inference speed and GPU memory usage among the connectors remain minimal.

Table 7: Runtime and memory comparison between different connector designs. The results show that ALIGN introduces negligible computational overhead compared to other connectors.

| Model | Samples | Avg Time (s) | Tokens/sec | GPU Memory (GB) |
|---|---|---|---|---|
| Llama-3.2-3B-MLP | 2500 | 0.161 | 118.3 | 10.9 |
| Llama-3.2-3B-Perceiver | 2500 | 0.140 | 135.1 | 10.9 |
| Llama-3.2-3B-Ovis | 2500 | 0.155 | 122.5 | 10.8 |
| Llama-3.2-3B-ALIGN | 2500 | 0.165 | 115.4 | 10.9 |

Overall, the empirical evidence confirms that the ALIGN connector achieves an effective balance between computational efficiency and performance. It introduces only a negligible increase in runtime and memory usage while providing substantial gains in overall accuracy.

## A.3 Pixel-Level Tasks Analysis

To rigorously evaluate the ability of vision-language models to integrate fine-grained visual and textual pixel-level cues, we test our model on the VCR benchmark [Zhang et al., 2024], which requires the model to recover partially occluded texts with pixel-level hints from the revealed parts of the text. This task challenges VLM's alignment of text and image in extreme situations. Current state-of-the-art models like GPT-4V OpenAI et al. [2023], Claude 3.5 Sonnet Anthropic [2024], and Llama-3.2 Dubey et al. [2024] significantly underperform humans on *hard* VCR task due to their inability to process subtle pixel-level cues in occluded text regions. These models frequently discard critical visual tokens during image tokenization on semantic priors, overlooking the interplay between partial character strokes and contextual visual scenes. To evaluate performance on VCR, we modify our Stage 3 SFT dataset composition by replacing the exclusive use of DocDownstream with a 5:1 blended ratio of DocDownstream and VCR training data. This adjustment enables direct evaluation of our architecture ALIGN's ability to leverage pixel-level character cues.

From the experimental outcomes, it is evident that ALIGNVLM consistently outperforms the MLP Connector Model across both easy and hard settings of the pixel-level VCR task (see Figure 5), with improvements ranging from 10.18% on the hard setting to 14.41% on the easy setting.

We provide a case study on VCR in Figure 6, featuring four representative examples. In Figure 6a, it is evident that the MLP connector model fails to capture semantic consistency as effectively as ALIGNVLM. The phrase "The commune first *census in written history in*" (where the words in italics are generated by the model while the rest are in the image) is not as semantically coherent as the phrase generated by ALIGN "The commune first *appears in written history in*".

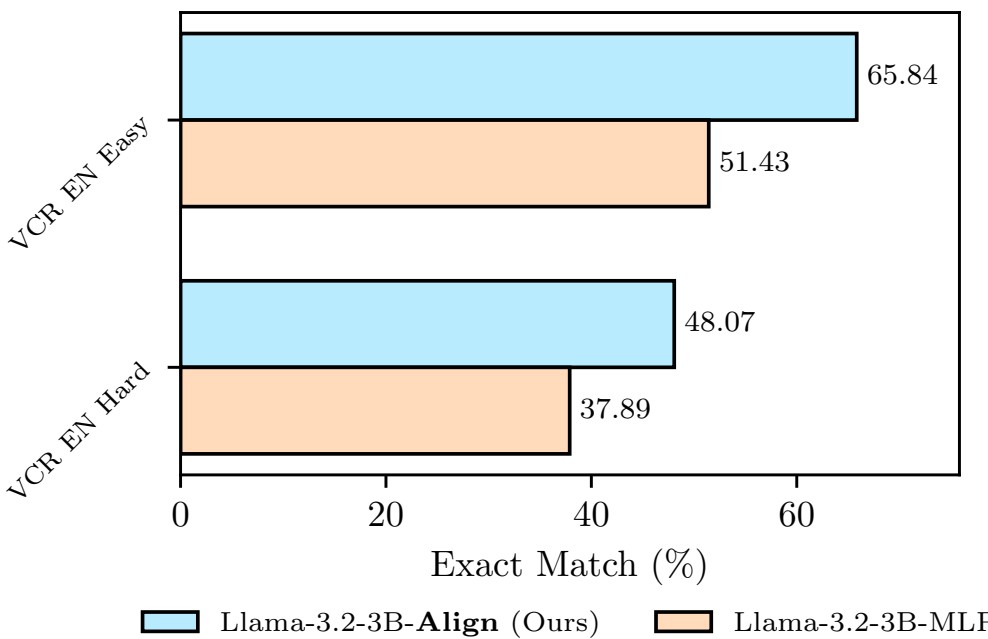

Figure 5: Comparison of Llama-3.2-3b-ALIGN and Llama-3.2-3B-MLP on the Easy and Hard VCR tasks.

Beyond the issue of semantic fluency, in Figure 6b we also observe that ALIGNVLM successfully identifies the uncovered portion of the letter "g" in "accounting" and uses it as a pixel-level hint to infer the correct word. In contrast, the MLP model fails to effectively attend to this crucial detail.

Figures 6c and 6d show examples where ALIGNVLM fails on the VCR task. These carefully picked instances show that our method mistakes names of landmarks with common words when the two are very similar. As seen in the examples, ALIGNVLM mistakes "Llanengan" for "Llanongan" and "Gorden" for "Garden". In both instances, the pairs differ by one character, indicating perhaps that ALIGNVLM tends to align vision representations to more common tokens in the vocabulary. One approach that would potentially mitigate such errors would be to train ALIGNVLM with more contextually-relevant data.

## A.4 Case Studies

In this section, we provide case studies for the experiments in Section 5.1. Specifically, we provide examples of our Llama-3.2-3B-ALIGN, and its counterpart model with alternative connectors Llama-3.2-3B-MLP and Llama-3.2-3B-Ovis on three different datasets: KLC [Stanisławek et al., 2021], DocVQA [Mathew et al., 2021b], and TextVQA [Singh et al., 2019]. The examples are shown in Figure 7, 8, and 9.

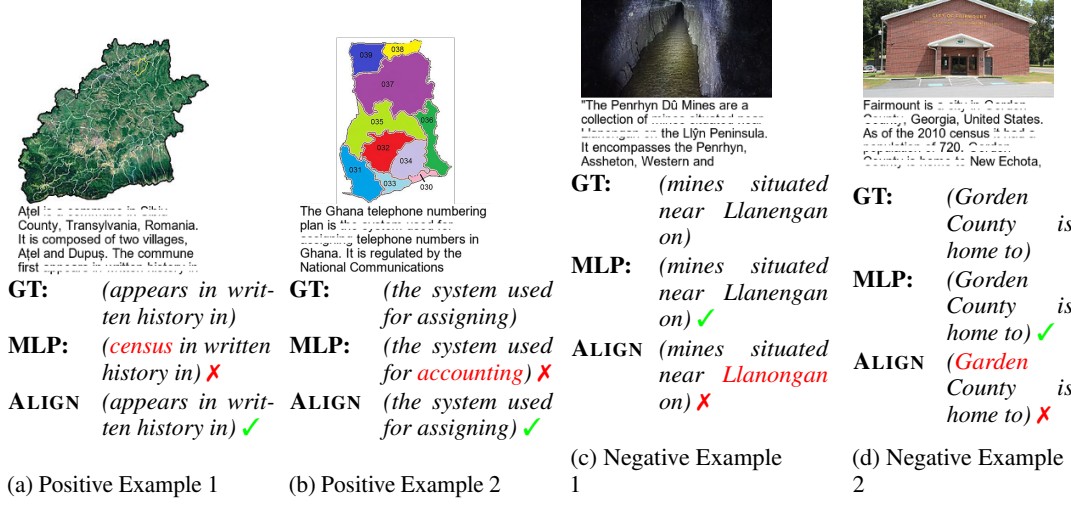

**GT:** *(appears in written history in)*

**MLP:** *(census in written history in)* ✗

**ALIGN:** *(appears in written history in)* ✓

(a) Positive Example 1

**GT:** *(the system used for assigning)*

**MLP:** *(the system used for accounting)* ✗

**ALIGN:** *(the system used for assigning)* ✓

(b) Positive Example 2

**GT:** *(mines situated near Llanengan on)*

**MLP:** *(mines situated near Llanengan on)* ✓

**ALIGN:** *(mines situated near Llanongan on)* ✗

(c) Negative Example 1

**GT:** *(Gorden County is home to)*

**MLP:** *(Gorden County is home to)* ✓

**ALIGN:** *(Garden County is home to)* ✗

(d) Negative Example 2

Figure 6: **Case Study for Pixel-Level Tasks.** We provide examples of our proposed **ALIGN** connector compared with a the Multi-Layer Perceptron (MLP) connector. The **ALIGN** connector tends to better map visual elements to common words. GT is the ground truth.

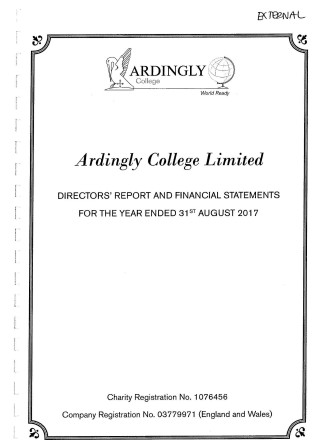

**Question:** What is the value for the charity name?
**GT:** *(Ardingly College Ltd.)*
**MLP:** *(Ardington College Ltd.)* ✗
**Ovis:** *(Ardington College Ltd.)* ✗
**ALIGN:** *(Ardingly College Ltd.)* ✓

(a) Positive Example #1

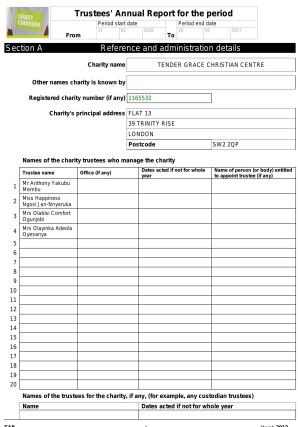

**Question:** What is the value for the address postcode?
**GT:** *(SW2 2QP)*
**MLP:** *(SW22 0PQ)* ✗
**Ovis:** *(SW2 2OP)* ✗
**ALIGN:** *(SW2 2QP)* ✓

(b) Positive Example #2

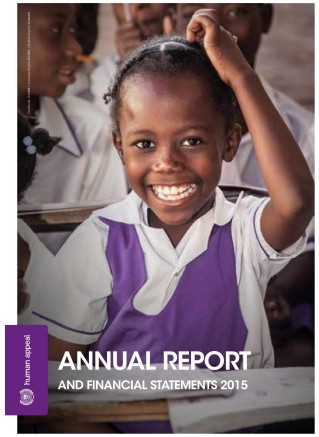

**Question:** What is the value for the charity name?
**GT:** *(Human Appeal)*
**MLP:** *(Humanitarian Agenda)* ✗
**Ovis:** *(Human Appeal)* ✓
**ALIGN:** *(Human Rightsappeal)* ✗

(c) Negative Example #1

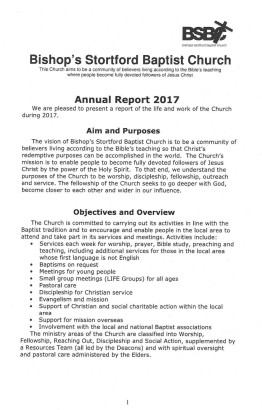

**Question:** What is the value for the post town address?
**GT:** *(Bishop's Stortford)*
**MLP:** *(Stortford)* ✗
**Ovis:** *(Bishop's Stortford)* ✓
**ALIGN:** *(Stortford)* ✗

(d) Negative Example #2

Figure 7: **Case Study for Connector Comparison on the KLC dataset [Stanisław et al., 2021].** We show four qualitative examples (including two correct and two incorrect examples) comparing Llama-3.2-3B-ALIGN to the same architecture with different connectors, Llama-3.2-3B-MLP and Llama-3.2-3B-Ovis. "GT" denotes the ground truth.

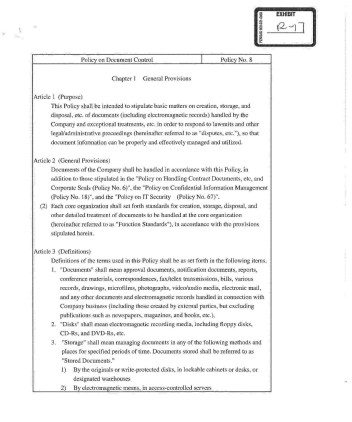

**Question:** What does the afternoon session begin on June 29?
**GT:** *(1:00)*
**MLP:** *(2:45)* ✗
**Ovis:** *(3:30)* ✗
**ALIGN:** *(1:00)* ✔

(a) Positive Example #1

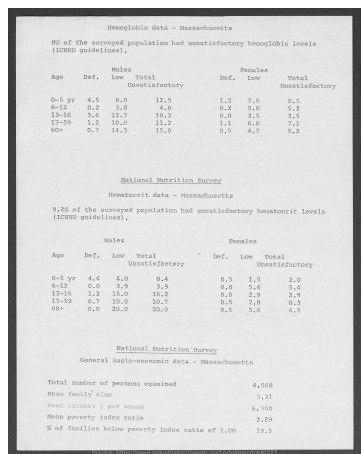

**Question:** What levels does the second table indicate?
**GT:** *(hematocrit data - Massachusetts)*
**MLP:** *(SATISFACTORY)* ✗
**Ovis:** *(Females)* ✗
**ALIGN:** *(hematocrit data - Massachusetts)* ✔

(b) Positive Example #2

**Question:** What type of policy is described in this document?
**GT:** *(Policy on Document Control)*
**MLP:** *(Policy on Document Control)* ✔
**Ovis:** *(General Provisions)* ✗
**ALIGN:** *(Document Control)* ✗

(c) Negative Example #1

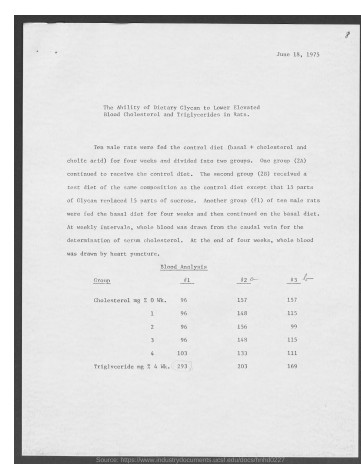

**Question:** What was the diet fed to the #1 group?
**GT:** *(basal diet)*
**MLP:** *(basel diet)* ✔
**Ovis:** *(Whole blood)* ✗
**ALIGN:** *(control diet)* ✗

(d) Negative Example #2

Figure 8: **Case Study for Connector Comparison on the DocVQA dataset [Mathew et al., 2021b].** We show four qualitative examples (including two correct and two incorrect examples) comparing Llama-3.2-3B-ALIGN to the same architecture with different connectors, Llama-3.2-3B-MLP and Llama-3.2-3B-Ovis. "GT" denotes the ground truth.

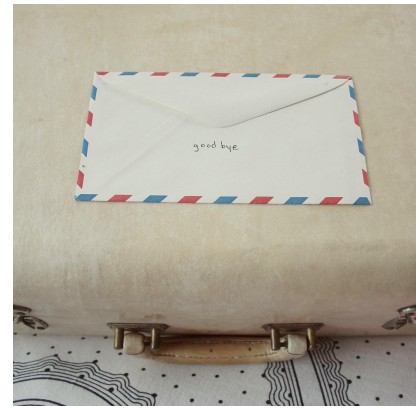

**Question:** What greeting is written on the letter?
**GT:** *(good bye)*
**MLP:** *(good)* ✗
**Ovis:** *(good buy)* ✗
**ALIGN:** *(good bye)* ✓

(a) Positive Example #1

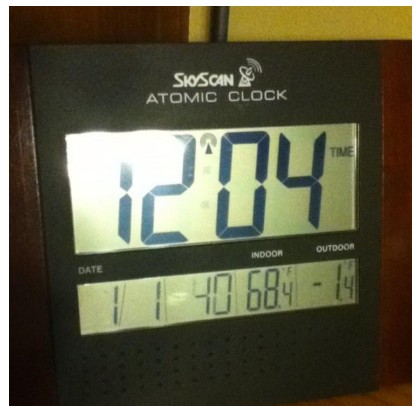

**Question:** What indoor temperature is shown?
**GT:** *(68.4)*
**MLP:** *(68 F)* ✗
**Ovis:** *(40.0)* ✗
**ALIGN:** *(68.4)* ✓

(b) Positive Example #2

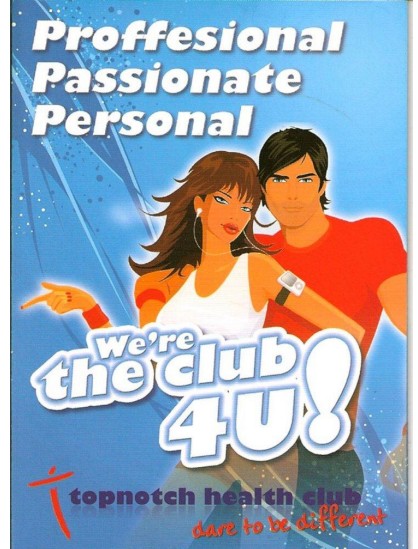

**Question:** What type of club is advertised?
**GT:** *(health club)*
**MLP:** *(topnote health club)* ✗
**Ovis:** *(health club)* ✓
**ALIGN:** *(professional passionate personal)* ✗

(c) Negative Example #1

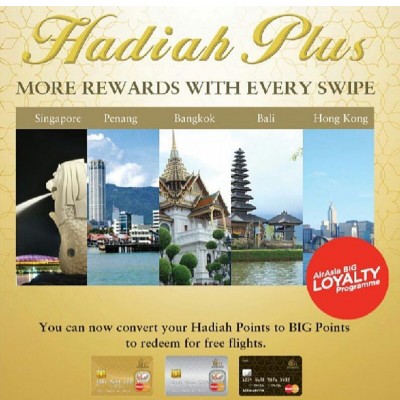

**Question:** What credit card is this?
**GT:** *(hadiah plus)*
**MLP:** *(hadiah plus)* ✓
**Ovis:** *(american big loyalty program)* ✗
**ALIGN:** *(hadia plus)* ✗

(d) Negative Example #2

Figure 9: **Case Study for Connector Comparison on the TextVQA dataset [Singh et al., 2019].** We show four qualitative examples (including two correct and two incorrect examples) comparing Llama-3.2-3B-ALIGN to the same architecture with different connectors, Llama-3.2-3B-MLP and Llama-3.2-3B-Ovis. "GT" denotes the ground truth.

