# OpenReview forum: "AlignVLM: Bridging Vision and Language Latent Spaces for Multimodal Document Understanding"
_NeurIPS.cc/2025/Conference — NeurIPS 2025 poster_

### Official Review · Reviewer_ZYJ3 · 2025-06-30

**Clarity:** 3
**Significance:** 2
**Originality:** 3
**Rating:** 4
**Confidence:** 4

**Summary:**

The paper introduces ALIGNVLM, a novel vision-language model architecture designed to improve the alignment between visual features and the latent space of Large Language Models (LLMs), particularly for document understanding. Instead of using traditional projection layers like MLPs, ALIGNVLM's connector maps visual features to a probability distribution over the LLM's entire vocabulary. It then generates visual embeddings as a weighted average of the LLM's existing text token embeddings. This approach constrains the visual inputs to lie within the convex hull of the text embedding space, which the authors argue prevents out-of-distribution features and enhances robustness. Through extensive experiments, the paper demonstrates that this method achieves state-of-the-art performance on various document-related benchmarks and shows superior resilience to noisy inputs compared to previous methods.

**Questions:**

Please, see weakness part.

**Ethical Concerns:**

["NO or VERY MINOR ethics concerns only"]

**Final Justification:**

The authors' comprehensive rebuttal, especially the new experiments against strong baselines, has addressed all my major criticisms. While I find the framing on the document domain slightly narrow given the method's general potential, the empirical results are now solid and convincing. The paper has been significantly improved, and I've raised my score.

**Limitations:**

The paper doesn't provide its societal impact. For the limitation, the paper only depicts that ALIGNVLM may be biased toward commonly used vocabs, in Appendix A.3.

**Quality:**

2

**Strengths And Weaknesses:**

### Strength
- The paper's most compelling and well-executed contribution is the noise robustness experiment (Section 5.5). By demonstrating a significantly smaller performance drop under noisy conditions compared to a standard MLP, the authors provide strong, quantifiable evidence for their core hypothesis. This finding clearly shows that constraining visual features to the LLM's latent space acts as an effective regularizer, which is a valuable and tangible result.


### Weakness
- Weak Justification for Scope and Generality
    - The paper's framing suffers from a lack of clear justification. While it predominantly focuses on document understanding, it fails to articulate a compelling reason why the ALIGN mechanism is uniquely suited for this domain. The argument that it should excel on documents because their visual content maps well to text tokens is implied but never explicitly made or proven.
    Furthermore, the paper's attempt to demonstrate broader applicability falls short. While results on general vision-language benchmarks are presented (Section 5.4), the comparison is limited to a basic MLP connector. This is an insufficient baseline. To make a convincing case for general-purpose utility, the ALIGN module needed to be benchmarked against more sophisticated and widely adopted general connectors, such as the OVIS, Perceiver Resampler or Q-Former. As it stands, the paper neither provides a strong rationale for its primary focus on documents nor does it rigorously validate its effectiveness as a general-purpose solution.
- Lack of Rigorous and Fair Experimental Comparison
    - The most significant weakness of this paper is its failure to properly situate ALIGNVLM within the competitive landscape of VLM connectors. The study's claims of superiority are unsubstantiated because it avoids the very comparisons needed to prove its merit. Specifically, two critical comparisons are missing:
        - **Omission of Key Architectural Baselines**: The paper neglects to perform a head-to-head architectural comparison against several state-of-the-art connectors, such as HoneyBee [A] or, most critically, the H-Reader module [B]. This is a fundamental flaw. To prove its value, a new architecture must be benchmarked directly against its most relevant competitors under identical training conditions.
        - **Invalid Justification via Unfair Benchmarking**: The paper attempts to sidestep a direct comparison with the H-Reader module by instead comparing its overall model performance to that of mplugDocOwl-1.5. This comparison is fundamentally unsound. There is a significant disparity in the training data used for ALIGNVLM versus what is typically used for SOTA models like mplugDocOwl-1.5, yet the paper makes no attempt to account for this crucial confounding variable. Therefore, claiming superiority based on this benchmark is misleading, as the performance difference cannot be confidently attributed to the ALIGNVLM architecture over the profound influence of the training data.
- Unconvincing Analysis
    - The qualitative analysis intended to offer insight into the model's inner workings is shallow and unconvincing. Figure 4, for instance, does not provide clear evidence that the model effectively distinguishes between text and background, nor does it show that text features are mapped to semantically meaningful embeddings. This superficial analysis fails to build a strong, evidence-backed narrative for how and why its method truly works, leaving its internal mechanisms poorly understood.

[A] Cha et al., Honeybee: Locality-enhanced Projector for Multimodal LLM, CVPR 2024

[B] Hu et al., mPLUG-DocOwl 1.5: Unified Structure Learning for OCR-free Document Understanding, arXiv 2024

---

> ### Author Rebuttal · Authors · 2025-07-31
>
> Thank you for your constructive feedback and acknowledging the robustness to noise of our Align connector. Below we address your concerns.
>
>
> # Concern 1: Scope and Additional Results on General Vision-Language Tasks
>
>
> Document understanding is challenging due to its inherent multimodal nature (integrating complex visual and textual information) and has numerous critical real-world industry applications. Thus, it serves as an ideal domain for evaluating our proposed AlignVLM approach. Our Align Connector maps visual features into convex combinations of the LLM’s text embeddings which are regions of the latent space that the LLM is inherently trained to interpret as text. As shown in Tables 2 and 3, the Align connector yields substantially larger improvements over existing connectors (e.g., MLP) on document benchmarks (Table 2) compared to general vision-language tasks (Table 3), supporting its design motivation.
>
>
> We agree with the reviewer that testing additional baselines on general vision-language tasks would further validate the generalizability of our method. To that end, we conducted ablation experiments using SigLIP-400M and Llama-3.2-3B with different connector designs, all trained on the LLaVA-Next dataset [1]. These models were trained under identical configurations: for pretraining, we used a learning rate of 1e-3, batch size of 64, and 1 epoch; for instruction tuning, we set the LLM and connector learning rate to 1e-5, vision encoder learning rate to 2e-6, batch size of 8, and 1 epoch. We evaluated the resulting models on standard general vision-language benchmarks including MMMU-dev, SeedBench, MMVet, POPE, and GQA. The results, summarized in the table below, demonstrate that the ALIGN connector outperforms all other connector variants on these general tasks on average, proving its effectiveness beyond the document domain.
>
> | Model                        | MMMU  | SeedBench | MMVet | POPE  | GQA   | Avg. |
> |-----------------------------|-------|-----------|-------|-------|-------|----------|
> | LLama-3.2-3B-MLP            | 33.33 | 58.54     | 31.14 | 87.35 | 57.62 | 53.59 |
> | LLama-3.2-3B-Perceiver      | 35.22 | 63.70     | 26.19 | 84.92 | 55.86 | 53.17 |
> | LLama-3.2-3B-Ovis           | 32.22 | 58.22     | 29.58 | 87.10 | 57.64 | 52.95
> | LLama-3.2-3B-HReducer       | 32.78 | 59.50     | 32.78 | 85.60 | 39.10 | 49.95 |
> | LLama-3.2-3B-HoneyBee (C-Abs)| 35.11 | 58.90     | 36.19 | 87.93 | 55.88 | 54.80 |
> | LLama-3.2-3B-Align          | 35.33 | 63.27     | 35.32 | 88.85 | 61.67 | 56.86 |
>
>
> # Concern 2: Comparison with HReducer and HoneyBee
>
> We would like to clarify that all models in the lower half of Table 1 (including mPLUG-DocOwl) were instruction-tuned on the same datasets (BigDocs-7.5M and DocDownstream). Table 1 is intended to provide a comparison between AlignVLM and state-of-the-art models at the full model level, under a consistent instruction-tuning regime.
>
> We also agree with the reviewer that a connector-level comparison with HReducer from mPLUG-DocOwl and HoneyBee (C-Abs) is necessary for a fair and direct architectural evaluation against our proposed connector (Align), similar to our comparisons with MLP, Perceiver Resampler, and Ovis in Table 2.
>
>
> To ensure a fair comparison under a compute-efficient yet controlled setup, we reused the same experimental configuration described in Concern 1 using SigLIP-400M and Llama-3.2-3B trained on the LLaVA-Next dataset with identical pretraining and instruction tuning settings. We evaluated all connectors, including HReducer and HoneyBee (C-Abs), on both document understanding benchmarks (DocVQA, InfoVQA, ChartQA, TextVQA) and general vision-language tasks (MMMU-dev, SeedBench, MMVet, POPE, GQA). As shown in the tables above (general vision tasks) and below (documents tasks), the Align connector consistently outperforms these strong baselines, highlighting its robustness across domains. Notably, HoneyBee is the second-best connector, likely due to its ability to preserve local visual context. We attribute ALIGN's superior performance to its unique design, which projects visual features into the convex hull of the LLM's text embeddings (a region in the latent space that the LLM is inherently pretrained to understand)
>
> | Model                         | DocVQA | InfoVQA | ChartQA | TextVQA |
> |------------------------------|--------|---------|---------|----------|
> | LLama-3.2-3B-MLP             | 42.11  | 19.93   | 48.44   | 51.97    |
> | LLama-3.2-3B-Perceiver       | 32.18  | 18.10   | 40.00   | 44.31    |
> | LLama-3.2-3B-Ovis            | 41.88  | 19.64   | 47.44   | 51.76    |
> | LLama-3.2-3B-HReducer        | 31.77  | 17.07   | 42.12   | 35.44    |
> | LLama-3.2-3B-HoneyBee (C-Abs)| 50.99  | 15.57   | 50.72   | 55.91    |
> | LLama-3.2-3B-Align           | 71.43  | 30.50   | 69.72   | 65.63    |
>
>
> # Concern 3: Additional Analysis of the Align Connector Internal Mechanism
>
> We conducted a deeper analysis of the token distributions described in Section 5.3. We observed that Align does not directly map visual patches to individual semantic tokens in a one-to-one manner. Instead, it computes a dense convex combination of the LLM text embeddings for each patch as we illustrated in Section 5.3. In addition, we noticed that Align consistently assigns high probabilities to approximately 3.4K tokens from the entire vocabulary, while assigning negligible probabilities (below 1e-6) to the remaining tokens. When we reduced the dimensionality using PCA and plotted the llm’s text embeddings in a 2D graph, we noticed that these 3.4K tokens densely and comprehensively span the latent space of all the LLM's text embeddings.
>
> To further validate this observation, we conducted additional evaluation experiments using only these 3.4K high-probability embeddings in the Align connector, removing the remaining embeddings entirely during evaluation. The results presented in the table below demonstrate negligible performance differences compared to using the complete set of embeddings (128K). This finding confirms that ALIGN effectively leverages these select embeddings to guide visual features into meaningful regions within the LLM's latent text space that the LLM can effectively understand. It also shows that the Align connector can benefit from further pruning to improve its efficiency significantly.
>
>
> | Model                         | DocVQA | InfoVQA | Deepform | KLC   | WTQ   | TabFact | ChartQA | TextVQA | TableVQA |
> |------------------------------|--------|---------|----------|-------|-------|---------|---------|----------|-----------|
> | AlignVLM-3B (3.4K tokens)     | 79.40  | 44.13   | 63.64    | 35.02 | 38.26 | 78.83   | 71.72   | 57.48    | 59.80     |
> | AlignVLM-3B (full)            | 79.63  | 44.53   | 63.49    | 35.25 | 38.59 | 78.51   | 71.88   | 57.38    | 60.10     |
>
> Unfortunately, we are not allowed to provide links to images in the rebuttal, but we will add these very interesting findings and figures in the revised manuscript to clarify the inner workings of the Align connector.
>
> Thank you for your thoughtful review. We hope our response has addressed your concerns and kindly ask you to **consider raising your score** to support the dissemination of our work within the ML community.
>
> [1] Llava-NeXT: Improved reasoning, OCR, and world knowledge

---

> > ### Comment · Reviewer_ZYJ3 · 2025-08-05
> > **Respond to the rebuttal**
> >
> > Thank you for the additional experiments, which address most of my concerns.
> >
> > However, the authors have not addressed my specific criticism of Figure 4. As other reviewers have also noted, Figure 4 fails to provide clear evidence that the model distinguishes between text and background or maps text features to semantically meaningful embeddings. I would appreciate the authors' thoughts on these concerns.

---

> > > ### Author Response · Authors · 2025-08-06
> > >
> > > Thank you for your response. We're glad our rebuttal has addressed most of your concerns. Below, we address your remaining point regarding Figure 4.
> > >
> > >
> > > We agree that Figure 4, in its current form, does not provide sufficient evidence to fully support our claim in lines 278–282 that the Align connector helps the model map visual features into semantically meaningful embeddings. Due to the very high-density nature of the token distribution (as shown in Figure 3, where the top token receives only a 0.0118 probability out of 128K tokens), inspecting only the top tokens in Equation 1 is not very informative. Since the final visual embedding is computed as a convex combination (weighted sum) over all 128K token embeddings (Equation 2), the resulting vector can differ significantly from any individual top token. This makes interpreting the one-to-one token mappings in Equation 1 potentially misleading.
> > >
> > >
> > > Instead, the strength of AlignVLM lies in constraining all visual features to lie within the convex hull of the LLM’s text embeddings, ensuring that the visual inputs to the LLM remain “in-distribution” similar to its original pretraining data inputs (text).  Other connectors like MLP do not impose such constraints and may generate out-of-distribution representations for this reason. This design of the Align connector significantly eases the learning process, especially in low-data regimes, as supported by our earlier results in the rebuttal and Table 2 in the paper. It also makes the connector more robust to noise as shown in Section 5.5.
> > >
> > > Additionally, as detailed in our original rebuttal, we observed that the model consistently assigns non-negligible probabilities to approximately 3.4K tokens out of the 128K-token vocabulary. Using only these 3.4K tokens (and excluding all others) yields comparable performance.  This shows that the model can utilize and combine only these 3.4K tokens to map the visual features into regions in the LLM’s convex hull that the LLM can interpret. When we reduced the dimensionality using PCA and plotted the LLM’s text embeddings in a 2D graph, we noticed that these 3.4K tokens densely and comprehensively span the 2D latent space of all the LLM's text embeddings.
> > >
> > >
> > > **We will revise our claims in lines 278–282 to reflect these new analyses instead which provide better insights into the internal mechanism of the Align connector. We will also replace Figure 4 with the PCA visualization of those 3.4K tokens over the LLM vocab.**
> > >
> > > Please let us know if you have any additional questions or feedback. We sincerely appreciate your constructive suggestions, which have improved the presentation of our work. Since we have addressed your key concerns (including comparisons with recent connector baselines such as HoneyBee and H-Reducer), we kindly ask you to consider revising your score to reflect your final evaluation of our work.

---

> > > > ### Author Response · Authors · 2025-08-08
> > > >
> > > > Dear Reviewer,
> > > >
> > > > Thank you for your constructive feedback, which has helped improve the quality of our work. As the rebuttal period concludes today, we wanted to kindly follow up to see if our previous response has addressed your last concern regarding Figure 4. We're happy to provide further clarification if needed.
> > > >
> > > > Since we have addressed your key concerns (including comparisons with recent connector baselines such as HoneyBee and H-Reducer), we kindly ask you to consider revising your score to reflect your final evaluation of our work.
> > > >
> > > > Best Regards,
> > > >
> > > > Authors

---

> > > > > ### Comment · Reviewer_ZYJ3 · 2025-08-09
> > > > >
> > > > > The authors' comprehensive rebuttal, especially the new experiments against strong baselines, has addressed most of my major criticisms. While I find the framing on the document domain slightly narrow given the method's general potential, the empirical results are now solid and convincing. The paper has been significantly improved, and I've raised my score.

---

### Official Review · Reviewer_tqc6 · 2025-07-01

**Clarity:** 3
**Significance:** 2
**Originality:** 3
**Rating:** 4
**Confidence:** 4

**Summary:**

This paper proposes ALIGNVLM, a novel vision-language model (VLM) framework that enhances cross-modal alignment between visual and textual representations, specifically targeting multimodal document understanding. The core contribution is the ALIGN connector, which avoids direct projection of visual features into the LLM’s embedding space. Instead, it represents each visual feature as a weighted average over the LLM's text embedding matrix, ensuring that visual inputs reside within the semantic space the LLM is pretrained to understand. The authors evaluate ALIGNVLM on a range of document-related tasks and compare it against both open-source and closed-source VLMs. Experiments demonstrate state-of-the-art performance, especially under limited training regimes, and provide in-depth analysis on robustness, alignment quality, and generalization to non-document V+L tasks.

**Questions:**

Low-resource regime analysis: The model is praised for efficiency, but no experiments are shown under low-resource settings (e.g., <1M pairs). Could the authors provide such analysis or comment on whether ALIGN still works with sparse data?
Claims of "parameter efficiency" (vs. deep fusion) lack quantifiable evidence (training/inference speed, memory).
Visual semantics leakage: Since ALIGN uses text embeddings to encode vision features, do you observe any semantic leakage (e.g., mapping to semantically plausible but incorrect tokens)?  Figure 4 suggests some alignment issues on white space – could these degrade QA performance?
No comparison with document-specialized models (e.g., LayoutLMv3, Pix2Struct). Is AlignVLM truly superior for structured content (tables/forms)?
Scalability to higher-resolution images: Your tiling approach supports up to 9 tiles.  How does the model scale (in performance and memory) when fed with dense layouts or longer document chains?

**Ethical Concerns:**

["NO or VERY MINOR ethics concerns only"]

**Final Justification:**

All of my concerns have been addressed, I have adjusted my score to borderline accept.

**Limitations:**

Yes. Authors partially discussed limitations (Sec 5.4) but should address:
How ALIGN would perform with LLMs that lack good vocabulary priors,
Possible misalignment in highly visual tasks (e.g., fashion, scene graph QA),
Societal risks such as hallucination when visual context is ambiguous.
Suggestion: Add an analysis/discussion section on transferability to less structured multimodal tasks and limitations of the convex-hull assumption in non-document domains.

**Paper Formatting Concerns:**

Radar Fig 1 axes with metric names (currently shows only scores).
The SOTA comparison in Table 1 is still not very clear.
Figures 3 and 4 could benefit from clearer axis labels and visual legends.

**Quality:**

2

**Strengths And Weaknesses:**

Strengths:
Quality – Strong empirical performance:
Extensive experiments across 9 document understanding benchmarks and several general V+L tasks.
ALIGNVLM outperforms both strong shallow fusion (e.g., MLP, Perceiver Resampler) and deep fusion models (e.g., LLaMA-3.2-11B), validating the method's effectiveness.
Novel alignment mechanism:Rather than learning new embedding spaces or large connector modules, ALIGN uses the LLM’s own vocabulary embeddings as a semantic anchor, preserving compatibility and reducing OOD risk.
Strong to noise:The connector shows significantly lower degradation under Gaussian noise than MLPs (−1.67% vs. −25.54%).
Parameter-efficient design:Outperforms larger models (e.g., ALIGN-3B vs. DocOwl-8B), highlighting efficient alignment without scaling up LLM parameters.
Well-organized and clearly written:Figures, tables, and appendix provide thorough support for claims; methodology is detailed and reproducible.

Weakness:
Limited theoretical insight:While the convex combination idea is intuitive, the paper lacks a formal theoretical justification or analysis of why this form of projection is optimal.
Data limitation disclosure needs clarification:Though the authors avoid using undisclosed instruction data, the actual differences in label coverage or image diversity between BigDocs vs. proprietary datasets are not deeply analyzed.
LLM dependency:The method’s reliance on the LLM's embedding space assumes high-quality pretrained embeddings; it's unclear how ALIGN performs with smaller or less capable LLMs.

---

> ### Author Rebuttal · Authors · 2025-07-31
>
> Thank you for your constructive feedback and acknowledging the novelty of our Align connector and its robustness to noise . We address your comments below.
>
> # Concern 1: Low-resource regime training
> We agree that experimenting with a low-data regime can further demonstrate ALIGN's efficiency. To address this, we conducted additional experiments using SigLIP-400M and Llama3.2-3B under a low-data regime by finetuning on the LLava-Next [1] dataset (779K data points). Specifically, we followed the LLava-Next recommended training configurations for both pretraining (freezing the llm and vision encoder, learning rate: 1e-3, batch size: 64, epochs: 1) and instruction tuning (LLM and connector learning rate: 1e-5, vision encoder: 2e-6, batch size: 8, epochs: 1).
>
> We evaluated models on both document understanding benchmarks (DocVQA, InfoVQA, ChartQA, TextVQA) and general vision-language tasks (MMMU-dev, SeedBench, MMVet, POPE, GQA). Our findings indicate that the Align connector significantly outperforms other connectors under the low-data regime, especially on document understanding tasks. This makes the Align connector very valuable in resource-constrained environments such as Academic labs.
>
>
> Document Understanding
>
> | Model                 | DocVQA | InfoVQA | ChartQA | TextVQA |
> |----------------------|--------|---------|---------|----------|
> | LLama-3.2-3B-MLP      | 42.11  | 19.93   | 48.44   | 51.97    |
> | LLama-3.2-3B-Perceiver| 32.18  | 18.10   | 40.00   | 44.31    |
> | LLama-3.2-3B-Ovis     | 41.88  | 19.64   | 47.44   | 51.76    |
> | LLama-3.2-3B-Align    | 71.43  | 30.50   | 69.72   | 65.63    |
>
> General VisionTasks
>
> | Model                 | MMMU  | SeedBench | MMVet | POPE  | GQA   |
> |----------------------|-------|-----------|-------|-------|-------|
> | LLama-3.2-3B-MLP      | 33.33 | 58.54     | 31.14 | 87.35 | 57.62 |
> | LLama-3.2-3B-Perceiver| 35.22 | 63.70     | 26.19 | 84.92 | 55.86 |
> | LLama-3.2-3B-Ovis     | 32.22 | 58.22     | 29.58 | 87.10 | 57.64 |
> | LLama-3.2-3B-Align    | 35.33 | 63.27     | 35.32 | 88.85 | 61.67 |
>
>
> # Concern 2: Runtime and Memory Comparisons with Deep Fusion Methods
> We compared Align against the deep fusion mechanism used in LLaMA-3.2 Vision [2]. We use the same backbone vision encoder and llm (SigLIP-400M, LLaMA-3.2-8B) to isolate the impact of the connector itself on inference runtime and memory usage. As shown in the table below, our Align connector reduces runtime and memory consumption, confirming our claim regarding its computational efficiency compared to deep fusion methods. We will include these details in the revised manuscript.
>
>
>
> | Connector           | # Params | Avg. Inference Time (s) | Tokens/s | GPU Memory (GB) |
> |---------------------|----------|--------------------------|----------|-----------------|
> | AlignVLM Connector  | 9B       | 34.12                    | 30.3     | 17.4            |
> | Llama 3.2 Connector | 10.3B    | 35.08                    | 29.4     | 20.0            |
>
>
>
> # Concern 3: Visual Semantic Leakage and Mappings
>
> To further understand the mapping process for Align, we conducted a deeper analysis of the token distributions described in Section 5.3. We observed that ALIGN does not directly map visual patches to individual semantic tokens in a one-to-one manner. Instead, it computes a dense convex combination of the LLM text embeddings for each patch as we illustrated in Section 5.3. In addition, we noticed that Align  consistently assigns high probabilities to approximately 3.4K tokens from the entire vocabulary, while assigning negligible probabilities (below 1e-6) to the remaining tokens. When we reduced the dimensionality using PCA and plotted all the llm text embeddings in a 2D graph, we noticed that these 3.4K tokens densely and comprehensively span the latent space of all the LLM's text embeddings.
>
> To further validate this observation, we conducted additional evaluation experiments using only these 3.4K high-probability embeddings in the Align connector, removing the remaining embeddings entirely during evaluation. The results presented in the table below demonstrate negligible performance differences compared to using the complete set of embeddings (128K). This finding confirms that Align effectively leverages and combines these select embeddings to guide visual features into meaningful regions within the LLM's latent text space that the LLM can effectively understand and interpret. It also shows that the Align connector can benefit from further pruning to improve its efficiency significantly.
>
> | Model                         | DocVQA | InfoVQA | Deepform | KLC   | WTQ   | TabFact | ChartQA | TextVQA | TableVQA |
> |------------------------------|--------|---------|----------|-------|-------|---------|---------|----------|-----------|
> | AlignVLM-3B (3.4K tokens)     | 79.40  | 44.13   | 63.64    | 35.02 | 38.26 | 78.83   | 71.72   | 57.48    | 59.80     |
> | AlignVLM-3B (full)            | 79.63  | 44.53   | 63.49    | 35.25 | 38.59 | 78.51   | 71.88   | 57.38    | 60.10     |
>
>
>
> Unfortunately, we are not allowed to provide links to images in the rebuttal, but we will add these very interesting findings and figures in the revised manuscript to clarify the inner workings of the Align connector.
>
> # Concern 4: Comparison with LayoutLMv3 and Pix2Struct.
>
> We believe that LayoutLMv3 and Pix2Struct are not directly comparable to our models due to fundamental architectural differences. LayoutLMv3 is an encoder-only model (similar to BERT), while Pix2Struct follows an encoder-decoder design. In contrast, AlignVLM is built on a vision-language architecture with an LLM decoder.
>
> Moreover, both LayoutLMv3 and Pix2Struct are relatively small in scale (typically 100M–300M parameters), whereas our models range from 1B to 8B parameters, making direct comparisons less meaningful and potentially misleading.
>
> We understand the importance of comparing AlignVLM with strong document-specific baselines. For this reason, Table 1 includes results for DocOwl-1.5, a recent and powerful document understanding model with a similar architecture and size. This offers a more appropriate and fair comparison. As shown in Table 1, AlignVLM consistently outperforms DocOwl1.5 across various benchmarks when instruction tuned on the same datasets, highlighting its effectiveness in the document domain.
>
> # Concern 5: Differences between BigDocs and Proprietary datasets
> Detailed datasets used by models like Qwen2-VL and InternVL remain undisclosed, making a completely unbiased comparison difficult. However, our deliberate selection of the BigDocs-7.5M and DocDownstream datasets was guided by their focus on document understanding which is the main focus and scope of our paper.
>
> Furthermore, open models (qwen2-vl, Intern-vl2) are anticipated to have been trained on significantly larger datasets (trillions of tokens and billions of images) and benefit from far more extensive GPU resources. We expect that this already places AlignVLM at a disadvantage, as larger-scale training typically leads to stronger generalization and performance improvements. Despite these constraints, AlignVLM remains competitive within the scope of multimodal document understanding. Our paper focuses explicitly on architectural improvements rather than extensive dataset curation.
>
> # Suggestion on Adding an analysis section on non-document tasks.
>
>
> Our paper already includes results on several widely used general vision-language benchmarks (MMMU, MMVet, POPE, GQA, and SeedBench) as presented in Section 5.4, demonstrating the generalizability of the Align connector beyond document understanding.
>
> We also agree with the reviewer that evaluating Align on additional diverse domains, such as fashion, scene understanding, or scene graph generation, would further strengthen the paper’s contribution. We appreciate this suggestion and will incorporate such analyses in the revised manuscript.
>
> Thank you for your thoughtful review. We hope our response has addressed your concerns and kindly ask you to **consider raising your score** to support the dissemination of our work within the ML community.
>
> [1] Llava-Next: Improved reasoning, OCR, and world knowledge
> [2] The Llama 3 Herd of Models

---

> > ### Comment · Reviewer_tqc6 · 2025-08-05
> > **Good Response**
> >
> > The authors have provided detailed and informative responses to my concerns, particularly by presenting additional experiments and analyses in several key areas. The comparisons of runtime and memory usage against deep fusion methods offer useful insights into the computational aspects of the Align connector. The low-resource regime training experiments shed light on its potential applicability in scenarios with limited data. Furthermore, the deeper analysis of the visual semantic mapping and the evaluation of selected high-probability tokens contribute to a better understanding of the model’s internal workings and possible efficiency improvements.
> >
> > Given these new results and clarifications, I have adjusted my score to borderline accept.

---

> > > ### Author Response · Authors · 2025-08-08
> > >
> > > Dear Reviewer,
> > >
> > > Thank you for your constructive feedback. We're glad our response addressed your concerns, and we appreciate your willingness to raise the score to a borderline accept.
> > >
> > > We just wanted to kindly note that the score still appears unchanged. Please feel free to update it when you submit your final rating.
> > >
> > > Best Regards,
> > >
> > > Authors

---

### Official Review · Reviewer_XUHv · 2025-07-01

**Clarity:** 4
**Significance:** 3
**Originality:** 3
**Rating:** 5
**Confidence:** 4

**Summary:**

In order to tackle with the problem of misalignment caused by connectors in VLMs, this paper proposes a new vision-text alignment technique, ALIGNVLM. It injects the embedding layer of LLM into the connector to generate a weighted average of LLM text embeddings in order to align vision features with LLM embedding space. The experiment results reveal a significant improvement on general document benchmarks brought by ALIGNVLM. Further analyses show that ALIGNVLM can also improve model's performance in general vision-language tasks including MMMU, SeedBench, MMVet, POPE, and GQA. What's more, ALIGNVLM also shows strong robustness to noise.

**Questions:**

1. Is the embedding layer in ALGNVLM freezed during training?
2. The injection of embedding layer will bring in more parameters. Can you show the comparison of training costs between ALIGNVLM and existing connnectors?

**Ethical Concerns:**

["NO or VERY MINOR ethics concerns only"]

**Final Justification:**

My questions and concerns have been adequatly addressed by the author rebuttal and I will keep my positive rating.

**Limitations:**

yes

**Quality:**

3

**Strengths And Weaknesses:**

strengths
1. The method is simple without huge change to the framework of VLMs but effective, and it provides a new insight in designing connectors for VLMs by injecting LLM embedding layer into them.
2. It can be applied to different domains including document comprehension and general vision-language tasks.
3. It shows stronger robustness to noises compared to models with different connectors.

weaknesses
1. ALIGNVLM is only applied to LLama in the experiments. Maybe more experiments with other VLMs should be conducted for a more solid conclusion.

---

> ### Author Rebuttal · Authors · 2025-07-31
>
> Thank you for acknowledging the effectiveness of the align connector and its robustness to noise. We address your concerns and questions below.
>
>
> # Concern 1: Experimentation with other LLMs
> We understand your interest in evaluating the Align Connector using other LLMs. However, due to computational constraints, we primarily experimented with the open-source SOTA Llama 3 models with different sizes (1B, 3B, and 8B parameters) to highlight Align’s scalability and consistent performance. Nonetheless, our Align module is LLM-agnostic and applicable in principle to any LLM. We have launched our training experiment with Qwen2.5-3B and will report the results in the camera-ready revised manuscript once it’s completed.
>
> # Question 1: Do we keep the embedding layers frozen during training?
> No, we train all weights of the Align connector in all stages, as detailed in Table 5 of the appendix. In initial experiments, freezing the LM head and the LLM embedding matrix in the Align connector resulted in unstable training. This is primarily due to the distribution shift in the LM head layer's inputs (in the Align Connector), from the previously encoded textual outputs of the LLM to the current visual features from our vision encoder.
>
> # Question 2: Runtime comparison between Connectors
>
> We acknowledge that the Align connector includes an additional LM head layer, which slightly increases the parameter count. However, this addition has a negligible impact on runtime efficiency due to its straightforward design, involving simple matrix multiplication operations (as shown in Equations 1 and 2 in the paper), rather than stacking multiple complex layers requiring sequential processing like the deep fusion methods.
>
> To empirically support this, we benchmarked the runtime and memory usage of models using different connectors (MLP, Align, Ovis, and Perceiver), corresponding to the experiments reported in Table 2 of our paper. The results below show minimal differences in inference speed and GPU memory usage between the connectors, despite the Align connector's notably superior performance as demonstrated previously in Table 2.
>
>
> | Model             | Samples | Avg Time (s) | Tokens/sec | GPU Memory (GB) |
> |------------------|---------|--------------|------------|-----------------|
> | mlp_module        | 2500    | 0.161        | 118.3      | 10.9            |
> | perceiver_module  | 2500    | 0.140        | 135.1      | 10.9            |
> | ovis_module       | 2500    | 0.155        | 122.5      | 10.8            |
> | align_module      | 2500    | 0.165        | 115.4      | 10.9            |
>
>
> Thank you for your thoughtful review. We hope our response has addressed your concerns.

---

> > ### Comment · Reviewer_XUHv · 2025-08-05
> > **Response to author rebuttal**
> >
> > I thank the authors for the detailed response to my comments. My questions and concerns have been adequatly addressed and I will keep my positive rating.

---

### Official Review · Reviewer_fkBd · 2025-07-02

**Clarity:** 3
**Significance:** 2
**Originality:** 3
**Rating:** 4
**Confidence:** 4

**Summary:**

The paper describes a new method to align visual and textual features in the context of multimodal vision-language models. The proposed method projects visual features as a weighted combination of textual embeddings of vocabulary tokens. This ensures that the projected visual features lie withing the convex hull of the textual embedding space. The proposed visual alignment is integrated into an state-of-the art Language Model and evaluated on Document Understanding tasks , showing improvement over other alignment methods, although the final performance is lower than current SoA.

**Questions:**

See above in strengths and weaknesses

**Ethical Concerns:**

["NO or VERY MINOR ethics concerns only"]

**Final Justification:**

The rebuttal have resolved some of my concerns, particularly those related to the design of the alignment method. I still have some concerns about the semantic interpretation of the aligned representation, but in light of the discussion with other reviewers, I agree with some of the positive comments made by other reviewers and I will raise my rating to borderline accept

**Limitations:**

Yes, limitations are addressed

**Paper Formatting Concerns:**

No formatting concerns

**Quality:**

2

**Strengths And Weaknesses:**

STRENGTHS
- The paper proposes a novel method for visual-text alignment, that tries to leverage prior semantics of the textual embedding space. Results show that this alignment can perform better than other existing alignment techniques.
- The paper is clear and well written,

WEAKNESSES
- I have some concerns about the design of the alignment technique. One of the problems of the existing methods (as claimed in the introduction) is that current methods can produce out-distribution and noisy representations. It is supposed that the proposed method would alleviate this problem. However, the first step in the proposed alignment method is a linear layer that projects the visual features into the textual space without any prior. Thus, this initial point that is used to get the final representation could also be noisy and out-of-distribution.
- One of the claims of the paper is that the representation obtained after the alignment can leverage semantic linguistic priors. However, the visualization of the token probability distribution in figure 3 and the predicted tokens in figure 4 raises several concerns about this claim. The dense probability distribution in figure 3 suggests that there is not a semantic differentiation according to the semantics of the visual token, but all textual tokens contribute similarly to the representation of visual tokens. In the same way, figure 4 seems to show that relevant textual tokens tend to concentrate on the most common vocabulary words, without any distinction according to the semantics of the visual token.
- Experimental results show a lower performance of the proposed alignment method with respect to SoA multimodal models, that also integrate some kind of alignment. When compared with models using the same data regime, all models decrease their performance with respect to the baseline version, which raises concerns about the data used for training the model. In table 1 I miss the comparison of the proposed method with exactly the same version of the baseline Llama model, to see exactly the effect of the alignment method.

---

> ### Author Rebuttal · Authors · 2025-07-31
>
> Thank you for your thoughtful comments and acknowledging the novelty of our Align idea. We address your comments below.
>
>
> # Concern 1: Design of Alignment Method
>
> The initial linear projection layer in Align is essential to match the dimensionality of the visual features with the LLM’s embedding space. This projection is effectively learned during training. The core innovation of the Align connector lies in the subsequent step, where visual features are mapped to a convex combination of the LLM’s text embeddings. This ensures that all visual inputs remain strictly within the convex hull of the LLM’s embedding latent space. This design naturally reduces the risk of noisy or out-of-distribution (OOD) representations. This is also supported by empirical results in Section 5.5; When Gaussian noise was added to the visual features, the Align connector exhibited only a 1.67% drop in performance, in contrast to a 25.54% drop observed with an MLP connector, highlighting the regularization benefits of our approach.
>
> # Concern 2: Align Connector Representation Learning
>
> We conducted a deeper analysis of the token distributions described in Section 5.3. We observed that Align does not directly map visual patches to individual semantic tokens in a one-to-one manner. Instead, it computes a dense convex combination of the LLM text embeddings for each patch as we illustrated in Section 5.3. In addition, we noticed that Align consistently assigns high probabilities to approximately 3.4K tokens from the entire vocabulary, while assigning negligible probabilities (below 1e-6) to the remaining tokens. When we reduced the dimensionality using PCA and plotted the embeddings in a 2D graph, we noticed that these 3.4K tokens densely and comprehensively span the latent space of all the LLM's text embeddings.
>
> To further validate this observation, we conducted additional evaluation experiments using only these 3.4K high-probability embeddings in the Align connector, removing the remaining embeddings entirely during evaluation. The results presented in the table below demonstrate negligible performance differences compared to using the complete set of embeddings (128K). This finding confirms that Align effectively leverages and combines these select embeddings to guide visual features into meaningful semantic regions within the LLM's latent text space that the LLM can effectively interpret. It also shows that the Align connector can benefit from further pruning to improve its efficiency.
>
> | Model                         | DocVQA | InfoVQA | Deepform | KLC   | WTQ   | TabFact | ChartQA | TextVQA | TableVQA |
> |------------------------------|--------|---------|----------|-------|-------|---------|---------|----------|-----------|
> | AlignVLM-3B (3.4K tokens)    | 79.40  | 44.13   | 63.64    | 35.02 | 38.26 | 78.83   | 71.72   | 57.48    | 59.80     |
> | AlignVLM-3B (full)           | 79.60  | 44.53   | 63.49    | 35.25 | 38.59 | 78.51   | 71.88   | 57.38    | 60.10     |
>
>
> Unfortunately, we are not allowed to provide links to images in the rebuttal, but we will add these very interesting findings/figures and revise our manuscript to reflect them. .
>
>
> # Concern 3: Slightly Lower Performance Compared to SOTA Models like Qwen2.5-VL-Instruct
>
> Detailed datasets used by models like Qwen2-VL and InternVL remain undisclosed, making a completely unbiased comparison difficult. However, our deliberate selection of the BigDocs-7.5M and DocDownstream datasets was guided by their focus on document understanding which is the main focus and scope of our paper.
>
> Furthermore, open models (qwen2-vl, Intern-vl2) are anticipated to have been trained on significantly larger datasets (trillions of tokens and billions of images) and benefit from far more extensive GPU resources. We expect that this already places AlignVLM at a disadvantage, as larger-scale training typically leads to stronger generalization and performance improvements. Despite these constraints, AlignVLM remains competitive within the scope of multimodal document understanding. Our paper focuses explicitly on architectural improvements rather than extensive dataset curation.
>
> To further justify our dataset choice and evaluate Align’s performance under low-data regime, we experimented with an alternative popular dataset, LLava-NeXT [1] which consist of 779K data points. We observed two critical findings: (1) Models trained on BigDocs-7.5M significantly outperform those trained on LLava-NeXT, and (2) Align significantly outperforms the MLP connector in a low-data regime, underscoring the strong inductive bias added through leveraging the LLM’s text embeddings for representing visual features. This makes the Align connector very valuable in resource-constrained environments such as Academic labs.
>
>
>
> | Model                           | DocVQA | InfoVQA | ChartQA | TextVQA | Average |
> |--------------------------------|--------|---------|---------|----------|---------|
> | LLama-3.2-3B-MLP (Llava Next)  | 42.11  | 19.93   | 48.44   | 51.97    | 40.61   |
> | LLama-3.2-3B-Align (Llava Next)| 71.43  | 30.50   | 69.72   | 65.63    | 59.32   |
> | LLama-3.2-3B-MLP (BigDocs)     | 71.46  | 37.56   | 66.48   | 53.56    | 57.26   |
> | LLama-3.2-3B-Align (BigDocs)   | 79.63  | 44.53   | 71.88   | 57.38    | 63.35   |
>
>
>
>
>
> # Concern 4: Performance of Comparable Llama Vision Models in Table 1
>
> Official Llama Vision models are limited to **two versions only**: Llama-3.2-90B (outside our scope) and Llama-3.2-11B which is reported in our manuscript (Table 1). Notably, the Llama-3.2-11B uses the exact LLM backbone (Llama-3.1-8B) as our AlignVLM-8B. The increased parameters from 8B to 11B in Llama-3.2-11B are primarily due to deep fusion connectors (additional cross-attention/feed-forward layers). The results in Table 1 shows that our AlignVLM-8B model achieves superior performance despite using a smaller connector, clearly demonstrating its efficiency and effectiveness compared to deep fusion methods.
>
>
> Thank you for your thoughtful review. We hope our response has addressed your concerns and kindly ask you to **consider raising your score** to support the dissemination of our work within the ML community.
>
> [1] Llava-Next: Improved reasoning, OCR, and world knowledge

---

> > ### Comment · Reviewer_fkBd · 2025-08-06
> >
> > Thank you for the detailed response that have resolved some of my concerns, particularly those related to the design of the alignment method and comparison with Llama. However, I still have some concerns about figures 3 and 4 and the semantic interpretation of the aligned representation and visual patches. The new results included in the rebuttal give some insight about it, but it does not fully show explain it.

---

> ### Author Response · Authors · 2025-08-06
>
> Thank you for your response. We're glad our rebuttal has addressed some of your concerns. Below, we address your remaining point regarding Figure 3 and 4.
>
> We agree that Figure 4, in its current form, does not provide sufficient evidence to fully support our claim in lines 278–282 that the Align connector helps the model map visual features into semantically meaningful embeddings. Due to the very high-density nature of the token distribution (as shown in Figure 3, where the top token receives only a 0.0118 probability out of 128K tokens), inspecting only the top tokens in Equation 1 is not very informative. Since the final visual embedding is computed as a convex combination (weighted sum) over all 128K token embeddings (Equation 2), the resulting vector can differ significantly from any individual top token. This makes interpreting the one-to-one token mappings in Equation 1 potentially misleading (Figure 4).
>
> Instead, we believe the strength of AlignVLM lies in constraining all visual features to lie within the convex hull of the LLM’s text embeddings, ensuring that the visual inputs to the LLM remain “in-distribution” similar to its original pretraining data inputs (text).  Other connectors like MLP do not impose such constraints and may generate out-of-distribution representations for this reason. This design of the Align connector significantly eases the learning process, especially in low-data regimes (llava-next dataset), as supported by our earlier results in the rebuttal and Table 2 in the paper. It also makes the connector more robust to noise as shown in Section 5.5.
>
> **We will revise our claims in lines 278–282 to reflect these new analyses (Previous Response to Concern 2) instead which provide better insights into the internal mechanism of the Align connector.**
>
> **We will also replace Figure 4 with the PCA visualization of the 3.4K tokens over the LLM vocab we discussed in the previous rebuttal response results (Concern 2).**
>
> Please let us know if you have any additional questions or feedback. We sincerely appreciate your constructive suggestions, which have improved the presentation of our work. If our responses have addressed your critical concerns, we kindly ask you to consider revising your score to reflect your final evaluation of our work.

---

### Official Review · Reviewer_T3uT · 2025-07-02

**Clarity:** 3
**Significance:** 3
**Originality:** 3
**Rating:** 4
**Confidence:** 4

**Summary:**

The paper introduces ALIGNVLM, a novel vision-language model architecture designed to improve multimodal document understanding. At the core of ALIGNVLM is a new connector module, ALIGN, which aligns visual features with a large language model (LLM) by mapping them to weighted combinations of pretrained LLM text embeddings. This approach constrains visual features to remain within the LLM’s latent space, addressing issues of out-of-distribution and noisy inputs common in previous shallow fusion methods like MLPs. Extensive experiments across multiple document understanding benchmarks (e.g., DocVQA, TableVQA, ChartQA) show that ALIGNVLM achieves state-of-the-art performance, outperforming both parameter-heavy deep fusion models and strong open-source instruction-tuned baselines, even when trained under the same data regime. The paper also presents robustness and generalization analyses, demonstrating the method’s effectiveness beyond document tasks and under noisy input conditions.

**Questions:**

1. How sensitive is ALIGN to the choice of the vision encoder? Could the authors provide ablations or analysis on whether stronger or weaker vision backbones (e.g., CLIP-ViT or smaller SigLIP variants) affect ALIGN’s performance or alignment robustness?

2. Can the authors offer a more detailed analysis or visualization of how ALIGN distributes visual features over text token embeddings? For instance, are the top tokens consistently interpretable or class-specific? How do these change across domains?

**Ethical Concerns:**

["NO or VERY MINOR ethics concerns only"]

**Final Justification:**

I will maintain my score since my concerns are addressed.

**Limitations:**

yes

**Quality:**

3

**Strengths And Weaknesses:**

**Strength**

1. The proposed ALIGN connector innovatively maps visual features to a convex combination of LLM text embeddings, ensuring the features remain in-distribution with respect to the LLM. This idea is simple, elegant, and grounded in strong inductive bias, avoiding ad hoc projection layers like MLPs.

2. ALIGNVLM consistently outperforms state-of-the-art shallow and deep fusion methods, including larger models like LLaMA 3.2-11B and DocOwl-8B, across a suite of diverse document understanding benchmarks (DocVQA, ChartQA, TableVQA, etc.).

3. The paper includes carefully controlled ablations comparing ALIGN with MLP, Perceiver Resampler, and Ovis under the same training regimes.

4. Although focused on document understanding, the method is tested on general V+L tasks (MMMU, POPE, GQA), showing ALIGN can generalize beyond the document domain.

**Weakness**

1. The paper uses SigLIP-400M, but does not analyze sensitivity to the vision encoder or explore whether ALIGN benefits from stronger backbones (e.g., CLIP ViT-G, SAM). This limits understanding of its scalability.

2. While the paper mentions encoder-free VLMs (like Fuyu-8B, EVE), it does not empirically compare ALIGNVLM against them, nor analyze trade-offs in removing the vision encoder altogether. This limits its positioning within the current landscape of lightweight VLM architectures.

3. The paper claims ALIGN is more efficient than deep fusion methods, but provides no concrete runtime, FLOPs, memory, or inference latency comparisons, which are critical in practical deployments.

---

> ### Author Rebuttal · Authors · 2025-07-31
>
> Thank you for your thoughtful comments and acknowledging that our Align idea is elegant and grounded in strong inductive bias. We address your comments below.
>
>
> # Concern 1: Experimenting with Other Vision Encoders
>
> In the paper, we focused on SigLip-400m due to its superior performance over CLIP [1]. We agree that experimenting with various vision encoders can further demonstrate ALIGN's generalizability. To address this, we conducted additional experiments using SigLIP-400M and CLIP vision encoders under a low-data regime setup for computational efficiency. Specifically, we finetune on the LLava-Next dataaset [2] and follow their recommended training configurations for both pretraining (freezing llm and vision encoder, learning rate: 1e-3, batch size: 64, epochs: 1) and instruction tuning (LLM and connector learning rate: 1e-5, vision encoder: 2e-6, batch size: 8, epochs: 1).
>
> We evaluated the models on both document understanding benchmarks (DocVQA, InfoVQA, ChartQA, TextVQA) and general vision-language tasks (MMMU-dev, MMVet, GQA). Our findings indicate that Align still produces strong and comparable performance despite using a weaker vision backbone. This proves that the Align connector generalizes well across different vision encoders.
>
>
> | Model                         | DocVQA | InfoVQA | ChartQA | TextVQA |
> |------------------------------|--------|---------|---------|----------|
> | LLama-3.2-3B–SigLIP-Align    | 71.43  | 30.50   | 69.72   | 65.63    |
> | LLama-3.2-3B–CLIP-Align      | 68.01  | 29.06   | 66.84   | 62.15    |
>
>
> | Model                         | MMMU  | MMVet | GQA   |
> |------------------------------|-------|-------|-------|
> | LLama-3.2-3B–SigLIP-Align    | 35.33 | 35.32 | 61.67 |
> | LLama-3.2-3B–CLIP-Align      | 33.67 | 33.99 | 59.95 |
>
>
> # Concern 2: Comparisons with Lightweight encoder-free VLMs (Fuyu-8B)
>
> We evaluated Fuyu-8B on the same document, understanding benchmarks and observed significantly limited performance compared to AlignVLM as shown in the table below. Although encoder-free models simplify architecture, they typically lack performance due to not leveraging pretrained visual components [3]. We will include these results in Table 1 and revise the discussions in our revised manuscript.
>
> | Model       | DocVQA | InfoVQA | Deepform | KLC   | WTQ   | TabFact | ChartQA | TextVQA | TableVQA |
> |-------------|--------|---------|----------|-------|-------|---------|---------|----------|-----------|
> | Fuyu-8B     | 48.97  | 23.09   | 4.78     | 6.63  | 14.55 | 47.91   | 44.36   | 46.02    | 15.49     |
> | AlignVLM-8B | 79.63  | 44.53   | 63.49    | 35.25 | 38.59 | 78.51   | 71.88   | 57.38    | 60.10     |
>
>
>
> # Concern 3: Runtime and Memory Comparisons with Deep Fusion Methods
> We compared Align against the deep fusion mechanism used in LLaMA-3.2 Vision [4]. We use the same backbone vision encoder and llm (SigLIP-400M, LLaMA-3.2-8B) to isolate the impact of the connector itself on inference runtime and memory usage. As shown in the table below, our Align connector reduces runtime and memory consumption, confirming our claim regarding its computational efficiency compared to deep fusion methods. We will include these details in the revised manuscript.
>
>
>
> | Connector           | # Params | Avg. Inference Time (s) | Tokens/s | GPU Memory (GB) |
> |---------------------|----------|--------------------------|----------|-----------------|
> | AlignVLM Connector  | 9B       | 34.12                    | 30.3     | 17.4            |
> | Llama 3.2 Connector | 10.3B    | 35.08                    | 29.4     | 20.0            |
>
> # Concern 4: Further Analysis of the Mapping of Visual Features over Embeddings.
>
> We conducted a deeper analysis of the token distributions described in Section 5.3. We observed that Align does not directly map visual patches to individual semantic tokens in a one-to-one manner. Instead, it computes a dense convex combination of the LLM text embeddings for each patch as we illustrated in Section 5.3. In addition, we noticed that Align consistently assigns high probabilities to approximately 3.4K tokens from the entire vocabulary, while assigning negligible probabilities (below 1e-6) to the remaining tokens. When we reduced the dimensionality using PCA and plotted the embeddings in a 2D graph, we noticed that these 3.4K tokens densely and comprehensively span the latent space of all the LLM's text embeddings.
>
> To further validate this observation, we conducted additional evaluation experiments using only these 3.4K high-probability embeddings in the Align connector, removing the remaining embeddings entirely during evaluation. The results presented in the table below demonstrate negligible performance differences compared to using the complete set of embeddings (128K). This finding confirms that Align effectively leverages these select embeddings to guide visual features into meaningful regions within the LLM's latent text space that the LLM can effectively interpret. It also shows that the Align connector can benefit from further pruning to improve its efficiency.
>
> | Model                         | DocVQA | InfoVQA | Deepform | KLC   | WTQ   | TabFact | ChartQA | TextVQA | TableVQA |
> |------------------------------|--------|---------|----------|-------|-------|---------|---------|----------|-----------|
> | AlignVLM-3B (3.4K tokens)    | 79.40  | 44.13   | 63.64    | 35.02 | 38.26 | 78.83   | 71.72   | 57.48    | 59.80     |
> | AlignVLM-3B (full)           | 79.60  | 44.53   | 63.49    | 35.25 | 38.59 | 78.51   | 71.88   | 57.38    | 60.10     |
>
> Unfortunately, we are not allowed to provide links to images in the rebuttal, but we will add these very interesting findings and figures in the revised manuscript.
>
> Thank you for your thoughtful review. We hope our response has addressed your concerns and kindly ask you to **consider raising your score** to support the dissemination of our work within the ML community.
>
>
> [1] Sigmoid Loss for Language Image Pre-Training (arxiv)
> [2] Llava-NeXT: Improved reasoning, OCR, and world knowledge
> [3] PaliGemma: A versatile 3B VLM for transfer
> [4] The Llama 3 Herd of Models

---

> > ### Comment · Reviewer_T3uT · 2025-08-04
> >
> > All my concerns are addressed and I will maintain my score.

---

### Comment · Area_Chair_Wua1 · 2025-08-02
**Discussion Phase**

Dear Authors and Reviewers,

I would like to thank the authors for providing detailed rebuttal messages.

I would like to encourage the **reviewers** to carefully read all other reviews and the author responses and engage in an open exchange with the authors. Please post your first response as soon as possible within the discussion time window, so there is time for back and forth discussion with the authors. All reviewers should respond to the authors, so that the authors know their rebuttal has been read.

Best regards, AC

---

### Decision · Program_Chairs · 2025-09-17

**Decision:**

Accept (poster)

**Comment:**

The paper introduces a connector that maps vision features to convex combinations of LLM token embeddings. The goal is to keep visual inputs in-distribution within the LLM’s text latent space. Experiments on document understanding (and some general tasks) show improved accuracy and robustness in comparison to standard connectors.

Reviewers highlight strong experimental gains using a simple and principled design, noise robustness  and low-data wins. The rebuttal adds controlled baselines and efficiency comparisons.

Initial concerns included baseline omissions and fairness questions. Reviewers saw interpretability figures as unconvincing but the authors promise to revise these. Some mentions of the theory being light and cross-LLM validation being limited.

During the discussion some reviewers raised their scores after new experiments and added low-data/efficiency analyses; one reviewer maintained caveats on analysis clarity.

Overall, I recommend to **accept** this paper to NeurIPS. Its clear idea with solid experimental evidence make it very suitable for presentation in my opinion.